

# Impact analysis of dynamical downscaling on the treatment of convection in a regional NWP model - COSMO: a case study during the passage of a very severe cyclonic storm "OCKHI"

Roshny S. [1], D. Bala Subrahamanyam [1,*], Anurose T. J. [2], and Radhika Ramachandran [1]

[1]Space Physics Laboratory, Vikram Sarabhai Space Centre, , Dept. of Space, Govt. of India, Indian Space Research Organization, , THIRUVANANTHAPURAM - 695 022, INDIA.
[2]Institute for Atmospheric and Environmental Sciences, , Goethe University, FRANKFURT 60438, Frankfurt a. Main, GERMANY

**Correspondence:** D. Bala Subrahamanyam (subrahamanyam@gmail.com)

**Abstract.** A significant source of uncertainty in Numerical Weather Prediction (NWP) models arises from the parameterization of sub-grid scale convection, whose inherent nature of complexity is amplified while applied to tropical regions where weather systems are controlled by many intricate factors. However, as the model resolution becomes finer, it is possible to switch off the convection parameterization, although it is still unclear at what resolution this can be achieved. Ambiguity arises due to
the inter-linking of various parameterization schemes within a model, and efficiency of one scheme depends on the output of another. In order to explore these issues, an intense convective episode with very heavy precipitation over the coastal Arabian Sea associated with the passage of OCKHI, one of the very severe cyclonic storms, is chosen as a case study. A set of distinct numerical simulations are carried out using Consortium for Small-scale Modelling (COSMO) to assess the direct and indirect impacts of dynamical downscaling on the treatment of convection. Results obtained from the present investigation indicate
dynamical downscaling together with switching off the convection parameterization could simulate the magnitudes of CAPE, one of the proxies for characterizing the occurrence of tropical convection, more realistically. But the downscaling did not improve the rainfall prediction, which were seen to deteriorate in the absence of convection parameterization.

# 1   Introduction

The failure of an NWP model in making accurate predictions of severe weather events is attributed to various factors such as inaccurate initial conditions, lack of data assimilation, poor representation of different sub-grid scale physical and atmospheric processes through parameterization, to name a few (Nikolina et al., 2014; Anurose, 2015; Kober et al., 2015; Clark et al., 2016). Among the various sub-grid scale physical processes which are parameterized, atmospheric convection has certain inherent features that distinguishes itself from other associated phenomena. For example, convection is one of the smallest and
most compact weather processes, and is directly responsible for the transportation of mass and energy from the surface to the



troposphere, hence it can affect the entire troposphere (Plant and Yano, 2015; Prein et al., 2015). It plays a major role in driving large-scale vertical circulations such as Hadley and Walker circulations by creating large horizontal gradients of latent heating (Stensrud, 2007; Arakawa and Jung, 2011). Convection process brings about atmospheric stability and is a driving factor in the vertical redistribution of energy (Tost et al., 2006; Martinez-Castro et al., 2016). Deep convection is considered as a significant

source of precipitation in many regions of the world and is known to be a leading cause of extreme events such as flash floods and landslides through heavy precipitation associated with mesoscale convective systems, squall lines, wave current interactions and tropical cyclones (Prein et al., 2015; Samiksha et al., 2017). Therefore, it is evident that the adequate representation of convection in NWP models is one of the essential components for accurate quantitative precipitation forecasts (Tost et al., 2006; Kuell and Bott, 2008; Plant, 2010). However, the process of convection is also inter-linked with complex cloud systems

which influence the atmospheric circulation through interaction with radiation. Methods for simulating convective precipitation have been a major focus during the evolution of NWP models over the past few decades. However, the progress in this direction has been relatively slow due to many ambiguities in both posing the problem as well as modelling the inter-linked processes in convective events (Vasconi et al., 2018). Further the different parameterization schemes employed in a model are inevitably linked with each other, and reducing the errors attributed to one scheme often requires improvements across the board to all

facets of the model (Stensrud, 2007).

In the area of weather forecasting, refinement in spatial grid resolution of the NWP models is apparently inter-linked with the advancement in computational infrastructure. With the availability of satellite-based meteorological observations in conjunction with the dense network of ground-based weather observations, spatial grid resolution of all the global as well as the regional NWP models is consistently becoming finer. As on today, most of the regional operational NWP models are configured

for a spatial grid resolution of a few kilometers. The initial and lateral boundary conditions for the regional NWP models are obtained by downscaling the coarse-grid meteorological data from the global models after incorporating the distinct features of the surface and model topography, which are generally available at far better spatial resolutions. To be more precise, a pure downscaling NWP system is one in which no data assimilation is carried out (apart from that done in the larger-scale global model from where it is downscaled). Even though the larger scale forecasts constraints the downscaled product, the process

of downscaling may still add considerable value in circumstances, when an intense convective event prevails over the study domain (Clark et al., 2016). At very fine grid resolution, treatment of sub-grid scale physical and atmospheric processes needs proper attention as the model attains inherent potential to explicitly resolve these processes. In this manner, dynamical downscaling of NWP models provide cloud resolving features, and the parameterization of convection itself tends to become an obsolete component within the NWP models. However, there is still an ambiguity within the NWP community on this aspect,

and a scientific debate is going on regarding the spatial grid resolution at which the parameterization of convection can be "SWITCHED OFF" in NWP models. The onus is on obtaining a balance with a sufficiently high resolution which gives more information about the convection and precipitation statistics without using a convection parameterization scheme, within the capabilities of the available computer resources.

With a view to addressing the above-mentioned aspects about the dynamical downscaling of an NWP model, here we present

a case study on the treatment of convection in a regional NWP model, namely - COSMO (Consortium for Small-scale Mod-



elling, more details on this model can be attained from the model website - http://www.cosmo-model.org/) during the passage of a "Very Severe Cyclonic Storm (VSCS)" - OCKHI over the central Arabian sea. OCKHI was one of the very rarest cyclonic storms whose genesis took place in the Comorin Sea and the storm subsequently traveled more than 2000 kms over the Arabian Sea before making its landfall on the western coastline of the Indian peninsula. In a climatological perspective, OCKHI attained

rapid intensification from a deep depression over the Comorin sea to a cyclonic storm within six hours itself (Subrahamanyam et al., 2018). In this research article, we focus our attention on the treatment of convection through a mass-flux parameterization scheme in COSMO and the impact of dynamical downscaling of model's spatial grid resolution on the precipitation fields during the passage of OCKHI cyclonic storm. The main objectives of this case study are broadly categorized in two folds: (1) What is the impact of dynamical downscaling of COSMO on the precipitation fields and its inter-relation with the mass-flux

convection parameterization scheme; and (2) How does the precipitation pattern simulated through COSMO vary with different initial conditions, typically corresponding to 48 h, 36 h, 24 h and 12 h prior to the intense phase of OCKHI cyclonic storm. Results obtained from the present study are discussed in view of downscaling, and the role of mass-flux convection parameterization scheme.

## 2 About the COSMO Model

The COSMO is a group of meteorological services from Germany, Greece, Italy, Poland, Romania, Russia, and Switzerland that pool their research and development resources in the field of regional NWP (Baldauf et al., 2011). The COSMO (formerly known as "Lokal-Modell" in Germany or "Alpine-Model" in Switzerland), a non-hydrostatic limited area atmospheric prediction model was initially developed at Deutscher Wetterdienst (German Weather Services) for operational NWP and various research applications on the meso$-\beta$ and meso$-\gamma$ scale and later in the framework of the COSMO consortium (Steppeler

et al., 2003; Buzzi, 2008; Baldauf et al., 2011). The three dimensional fully elastic and non-hydrostatic atmospheric equations in this model are based on the primitive thermo-hydrodynamical equations and are solved numerically with second- or third-order finite differences on an Arakawa-C grid system. The solution of the dynamical governing equations is based on the Runge-Kutta scheme with a prescribed time step chosen by the user. The lowest vertical level of the 50 terrain following layers is placed at 10 m above the local topography and a generalised terrain following height co-ordinate system is adopted

for the definition of vertical grid points. The prognostic variables in this model include horizontal and vertical Cartesian wind components, pressure perturbation, temperature, specific humidity, cloud water content, cloud ice content, turbulent kinetic energy, specific water content of rain, snow and graupel, whereas the total air density as well as the precipitation fluxes of rain and snow are treated as diagnostic variables (Anurose and Subrahamanyam, 2015; Anurose, 2015). The COSMO model includes parameterization for radiative transfer, cloud micro-physics, sub-grid scale turbulence, and convection. The model

also includes parameterization for the ground heat and water transport and the land-atmosphere interactions. The formation of precipitation fields is described by a bulk micro-physics parameterization including water vapour, cloud water and ice, rain and snow with a fully prognostic treatment of precipitation, i.e., three dimensional transport of rain, and snow is directly calculated (Milelli et al., 2008). Condensation and evaporation are parameterized by saturation adjustment whereas depositional growth



and sublimation of cloud ice are estimated using an explicit non-equilibrium growth equation. Sub-grid scale cloudiness for radiation calculations is parameterized by an empirical function depending on relative humidity, ice content and height. Treatment of moist convection is based on a mass-flux parameterization scheme suggested by Tiedtke. Time integration in COSMO uses a second order leapfrog (horizontally explicit, vertically implicit) time-split integration approach. Different schemes of

initialisation are available in the COSMO model, wherein the initial and lateral boundary conditions are fed to the model through a tailored sub-set of global model's analysis and forecast fields.

## 2.1  Parameterization of Convection in NWP model COSMO

The basic role of a convection parameterization scheme lies on the mathematical representation of time-evolution of direct and indirect impacts of convective processes on the prognostic variables under the conserved framework of mass, momentum

and energy. Different schemes of parameterization for convective processes can be conceptualised in many different ways, but ultimately these schemes allow numerical estimation of the collective effects of convective clouds in a model column in terms of the grid-scale resolved variables. Since the convective processes operate on horizontal scales which are much smaller than those resolved by mesoscale and regional NWP models, the basic effects of moist convection on the prognostic variables must be carefully resolved through the convection parameterization schemes. One of the challenging problems in

the area of convection parameterization is to deal with the diabatic heating due to the release of latent heat resulting from cloud condensation and from the formation and evaporation of precipitation and the vertical transport of heat, moisture and momentum in cumulus updrafts and downdrafts as well as in the regions with compensating downwards motions, which in turn interact with the cumulus clouds by lateral exchange processes (Buzzi, 2008). Based on the vertical extent of the atmospheric forcing that controls convective processes, Mapes (1997) has classified the convection parameterization schemes into two broad

categories, namely: (1) Deep layer control schemes; and (2) Low level control schemes. Deep layer control schemes tie the creation of the Convectively Available Potential Energy (CAPE) by large-scale processes to the development of convection where convection consumes the CAPE that is created. On the other hand, Low level control schemes tie the development of convection to the initiation processes by which Convective Inhibition (CIN) is removed. In addition, CAPE can be generated and stored for long periods before it is consumed by the scheme.

COSMO model currently offers two schemes of convection parameterization based on the Low level control schemes: (i) the default scheme based on the closure assumptions reported by Tiedtke in 1989 (Tiedtke, 1989) (hereafter referred to as the Tiedtke scheme); and (ii) an alternative scheme based on the assumptions by Bechtold et al. (2001, 2014), (hereafter referred to as Bechtold Scheme) initially implemented for the ECMWF (European Centre for Medium-range Weather Forecast) global model. These schemes adopt a mass-flux approach to represent moist convection in numerical models and discriminate shallow,

mid-level and deep convection from each other by different closure hypotheses. The main differences in these two schemes lies in the closure assumptions. In the case of deep convection, an equilibrium type of closure is applied in the Tiedtke scheme by imposing a moisture balance for the sub-cloud layer in such a way that the vertically integrated specific humidity is preserved in the presence of grid-scale, turbulent and convective transports (Kuo and Raymond, 1980; Tiedtke, 1989). On the other hand, an equilibrium between the large-scale and boundary-layer forcing (generating CAPE) and convection (reducing the CAPE) is



assumed in the Bechtold formulation (Bechtold et al., 2001). Thus, in the Tiedtke scheme, only one type of convection may be present at a grid point at a time and hence layered convection cannot be described. Tost et al. (2006) provides a detailed review on the uncertainties in the model forecasts arising from different schemes of parameterization for atmospheric convection in a General Circulation Model (GCM). Liu et al. (2005) have conducted different sensitivity tests using several versions of the

Tiedtke convection scheme in the National Center for Atmospheric Research (NCAR) Community Atmosphere Model (CAM2) for investigating different aspects of the Boreal winter Madden-Julian Oscillations (MJO). Based on their study, it was found that, the Tiedtke convection scheme gives an improved mean state, intra-seasonal variability, space-time power spectra, and eastward propagation compared to the standard version of the model. In the global as well as mesoscale regional atmospheric models, the Tiedtke convection parameterization scheme is widely accepted and extensively used for the prediction of severe

weather events (Liu et al., 2005; Akkermans et al., 2012; Osuri et al., 2012). Recent developments in the area of dynamical downscaling and parameterization of convection show that the realism of convection-permitting models can provide improved forecasts for subjective guidance on convection and eventually can improve rainfall forecasting skill compared to the traditional approach wherein the sub-grid scale convection is treated through some scheme of parameterization.

In the present study, COSMO model is initialised with the initial and lateral boundary conditions extracted from the analysis

and forecast fields of ICON (ICOsahedral Nonhydrostatic, a German global model). Further technical details on the COSMO model configuration used in the present study are described in Table A1. More details on the functioning of COSMO are available on the model's website and other published literature (Steppeler et al., 2003; Buzzi, 2008; Baldauf et al., 2011; Anurose and Subrahamanyam, 2015; Anurose, 2015).

## 3   Data

### 3.1   Model Simulations

The research work is focused on an intense rainfall episode associated with the passage of OCKHI - a VSCS over the central Arabian sea. Genesis of this cyclonic storm was observed over the Comorin sea in the last week of November 2017 and afterwards the storm made north-westward progression and subsequently attained the status of a VSCS on the afternoon of $01^{st}$ December 2017. The primary objective of this case study is focused on the treatment of convection in COSMO and

the impact of dynamical downscaling on the precipitation fields over the central Arabian sea, which received very intense rainfall during the progression of OCKHI. Hence, a geographical region of about 1000 km x 1000 km centered around the coastal Arabian sea between 66°E to 75°E longitudinal band and extending from 7°N to 16°N latitudes,is configured as the model domain. The analysis fields corresponding to 00 and 12 UTC of ICON global model for a period of 9 days from $28^{th}$ November 2017 to $06^{th}$ December 2017 were utilised as the initial conditions for each distinct simulations, and forecast fields

for +48 h were generated from all individual simulations. Different meteorological parameters extracted from the forecasts fields corresponding to above simulations form the COSMO model database.



## 3.2 Observational Fields

For the validation of model-simulated forecast fields, we have made use of the ECMWF Reanalysis (ERA) - Interim fields, which are standard reanalysis fields and are very widely used in the NWP applications (Dee et al., 2011). These observational reanalysis fields are available at a grid resolution of $0.70°$ over the tropics. Among various parameters available in the reanalysis

fields, we have chosen CAPE and accumulated precipitation fields as two important variables for characterising convective event. In addition to the reanalysis fields, we have also made use of the inferred track information from the internal report published by the India Meteorological Department (IMD, http://www.imd.gov.in) on the time progression of OCKHI including the exact location of the eye of the storm.

## 4   Numerical Experiments in the COSMO Model

In the present study, we have extracted the ICON analysis fields corresponding to 00 and 12 UTC of $01^{st}$ and $02^{nd}$ December 2017 for providing the initial conditions to COSMO. Different atmospheric and meteorological fields of the ICON German global model are available at a spatial grid resolution of $0.13°$ (Zangl et al., 2015; Schattler et al., 2016; Heinze et al., 2017). COSMO model offers an interface program, namely *"int2lm"* which takes the surface-layer and topographical constant fields, together with the the varying atmospheric fields of ICON for generation of initial conditions for the COSMO model domain at

a fine resolution chosen by the COSMO user. The basic purpose of this interpolation program is to smoothen the topographical features over the model domain and make use of fine-features of surface characteristics with a special emphasis on the boundaries from a coarse-grid resolution to a fine-grid resolution. In the present study, we have chosen Tiedtke scheme of convection, which is a well-tested scheme and is also used as the default parameterization scheme of convection in COSMO model (Liu et al., 2005; Akkermans et al., 2012).

A total of three unique numerical simulation experiments are designed and carried out in the present study for addressing the impact analysis of downscaling on the treatment of convection in COSMO model. These numerical experiments are schematically depicted in Figure 1. Below, we explain the details about these numerical experiments (also refer to Table A1):

1. **Control Parameterized Convection (CPC) Simulations:**

   In the first set of simulations (hereafter referred to as the CPC simulations), COSMO model is configured for a spatial grid

25       resolution of $0.0625°$( = approximately 7 kms), and the necessary constant and varying surface/atmospheric variables are downscaled from coarse grid (~$0.13°$) fields of ICON to a fine grid (~$0.0625°$) of COSMO for the generation of initial and lateral boundary conditions. For CPC simulations, convective processes are parameterized under the framework of Tiedtke convection parameterization scheme. For capturing the severe convective episode and associated rainfall fields observed on 00 UTC of $03^{rd}$ December 2017, a total of four distinct Control simulations (namely: CPC-48, CPC-36,

30       CPC-24, and CPC-12) are carried out. The CPC-48 simulation is carried out with the initial conditions corresponding to 00 UTC of $01^{st}$ December 2017, 48 h prior to the occurrence of actual convective episode. Similarly, CPC-36, CPC-24 and CPC-12 control simulations makes use of the initial conditions corresponding to 12 UTC of $01^{st}$ December 2017,



UTC of $02^{nd}$ December 2017 and 12 UTC $02^{nd}$ December 2017 respectively. In other words, the CPC-48, CPC-36, CPC-24 and CPC-12 control simulations provide a lead time of 48 h, 36 h, 24 h and 12 h respectively to the COSMO model towards prediction of actual episode that took place on 00 UTC of $03^{rd}$ December 2017.

2. **Down-scaled Parameterized Convection (DPC) Simulations:**

The second set of simulations (hereafter referred to as the DPC simulations) are almost similar to the CPC simulations, however in this set of simulations the initial and lateral boundary conditions to the COSMO model are provided by downscaling the analysis and forecast fields to a spatial grid resolution of 0.025°. Here as well, the convective processes are treated under the framework of Tiedtke convection parameterization scheme. Following the same analogy of nomenclature, the DPC-48, DPC-36, DPC-24 and DPC-12 simulations are named for the COSMO simulations with a lead time of 48 h, 36 h, 24 h and 12 h respectively. With reference to the original ICON analysis and forecast fields (available at 0.13° spatial grid resolution), dynamical downscaling in this set of simulations yields a fine horizontal mesh at a grid resolution of 0.025°, almost five times finer than that of the ICON.

3. **Downscaled with No Parameterization of Convection (DNC)Simulations:**

In this set of simulations (hereafter referred to as the DNC simulations), the convective processes are treated directly, and the provision for parameterization of convection is switched off in the namelist of configuration files dealing with physical parameterization schemes within the COSMO. For fine-grid spatial resolutions, it is recommended to switch off the moist convection parameterization scheme at spatial grid resolutions finer than 3 km, and the convective processes at such finer resolutions must be treated directly. Similar to the previous two experiments, we have adopted the same nomenclature analogy, and the DNC-48, DNC-36, DNC-24 and DNC-12 simulations make use of the initial conditions corresponding to 48 h, 36 h, 24 h and 12 h prior to 00 UTC of $03^{rd}$ December 2017 respectively.

The above mentioned numerical experiments provide detailed insights on two aspects: (i) role of Tiedtke convection parameterization scheme on the precipitation forecast fields and (ii) performance of COSMO model with different initial conditions with a special emphasis on lead time requirements for forecasting severe weather events.

## 5 Results and Discussion

### 5.1 Intense Convective Precipitation Episode over the coastal Arabian Sea

OCKHI was one of the rarest cyclonic storms with its total residence time of 162 hours during which it traveled a distance of about 2538 kms over the Arabian sea (Subrahamanyam et al., 2018). The category of cyclone, its intensity, and the location of the eye are also taken from the IMD report for validation of COSMO simulated fields. This cyclonic storm was categorized as a VSCS on the afternoon of $01^{st}$ December 2017 (Subrahamanyam et al., 2018). After attaining the status of VSCS, a broad region of the central Arabian sea received intense rainfall between $01^{st}$ December 2017 to $05^{th}$ December 2017. Due to its long residency period and associated intense rainfall over the central Arabian sea, this cyclonic event becomes an ideal case





for investigation on the treatment of convection in an NWP model. We have chosen a geographical area over the Arabian sea as the model domain centered over the region covering convective event from $02^{nd}$ December to $03^{rd}$ December 2017 (see Figure 2). The CAPE is considered as a proxy for identifying the occurrence of convection, hence the spatial fields of CAPE corresponding to 00 UTC of $03^{rd}$ December 2017, when the central Arabian sea was under the influence of intense rainfall

due to OCKHI, are depicted in Figure 2. Large values of CAPE (>2000 J/kg) are clearly seen around the eye of the cyclonic storm. The progression of OCKHI cyclonic storm from its formation as a depression on the early morning of 29th November 2017 to its final dissipation and landfall on $06^{th}$ December 2017 are also marked along the trajectory. The intense convective conditions prevailing over the Arabian sea for 00 UTC of $03^{rd}$ December 2017 are chosen as the target meteorological fields which are simulated through COSMO based on different initial conditions. After the genesis of OCKHI cyclonic storm over

the Comorin sea, the storm gained intensity and moved north to north-westward and finally became a VSCS on the afternoon of $01^{st}$ December 2017. During its passage over the coastal Arabian sea from 29th November 2017 to $03^{rd}$ December 2017, the storm moved north-westward, resulting in heavy (64.5 mm to 115.5 mm) to very heavy (>115.5 mm) precipitation over the Lakshadweep islands. On $02^{nd}$ December 2017, IMD issued very heavy rainfall alerts of about 200 mm or more for the Lakshadweep islands. For this period, the eye of the cyclonic storm was centered quite close to Lakshadweep islands.

Subsequently, the cyclonic storm moved north-westward on $03^{rd}$ December 2017 but it still remained as a VSCS, and the Arabian sea experienced very heavy precipitation (>200 mm). Based on the intensity of convection, and category of OCKHI on these days, we have identified the severe convective precipitation episode from 00 UTC of $02^{nd}$ December 2017 to 00 UTC $03^{rd}$ December 2017 over the Lakshadweep islands and surrounding Arabian sea as the primary case for investigation of the treatment of convection in COSMO model. It may also be noted that, the accumulated precipitation fields obtained from the

ECMWF reanalysis for $02^{nd}$ to $03^{rd}$ December 2017 also indicated rainfall exceeding 200 mm within 24 h. Subrahamanyam et al. (2018) have reported simulated location of the eye of cyclone at a place which was almost 40.1 km away from the observed eye of the cyclone corresponding to 00 UTC of $02^{nd}$ December 2017. This deviation was about 41.7 km for 00 UTC of $03^{rd}$ December 2017. Figure 3 depicts 24 h accumulated rainfall over the model domain for $03^{rd}$ December 2017 as seen in (a) ECMWF Reanalysis fields, together with the concurrent +48 hours and +24 hours advanced COSMO forecasts [(b)

and (c) respectively]. The COSMO model forecasts as well as the ECMWF Reanalysis indicated intense rainfall (>250 mm) around the trajectory of OCKHI. Since the ECMWF Reanalysis fields are available at a coarse resolution of 0.70°, we don't see any fine-features as seen in the COSMO forecasts. In addition to this, the magnitude of rainfall seen in ECMWF fields were larger than that in the COSMO forecasts. Also, the horizontal wind fields (not shown here) for both the COSMO forecasts and ECMWF Reanalysis fields were close to each other, and the eye of the cyclone inferred from both these datasets were within a

mean deviation of about 46.17 kms. In the subsequent sections, our focus is on simulation of the intense convective episode and associated rainfall fields which took place during a period of 24 h spanning from 00 UTC of $02^{nd}$ December to $03^{rd}$ December 2017.





### 5.1.1 Simulation of CAPE for 00 UTC of $03^{rd}$ December 2017

Spatial maps of CAPE obtained from three set of simulations were carefully examined for understanding the intensity of convection as well as its spatial coverage over the study domain. CAPE over a given region is one of the stability indices used by the meteorologists to infer the potential of lower atmosphere to yield severe convective events and associated precipitation.

Atmospheric conditions with background fields of CAPE exceeding 2500 J/kg are considered to be proactive for the occurrence of thunderstorms (Uma and Das, in press, 2018). Figure 4 (a) depicts the spatial maps of CAPE extracted from 00 UTC of $03^{rd}$ December 2017 from the ERA Interim Reanalysis fields. The progression of OCKHI storm is also overlaid on the figure for quick reference. The coastal Arabian sea experienced severe convective activities as the storm made its passage and large values of CAPE (>2500 J/Kg) were observed in the proximity of the eye of the cyclone, which was reported at a location

69.15°E, 11.82°N on 00 UTC of $03^{rd}$ December 2017, and the category of storm was retained as VSCS. The intense convective activities inferred from the CAPE values of ECMWF fields were confined to 150 km x 75 km as clearly revealed in Figure 4 (a). COSMO simulations for this period also indicated the presence of a VSCS over the coastal Arabian sea, however the eye was approximately 41 kms away from the actual observations. The ECMWF fields were almost off by more than 100 kms from the actual position of cyclonic storm as reported by IMD. Figures 4 (b), (c), and (d) represent the 48 hours advanced CAPE

forecasts by COSMO for three different numerical experiments CPC, DPC and DNC respectively. CPC-48 predicts intense convective activities (CAPE >2000 J/Kg) for a widespread region including the spiral arms of the cyclonic storm (Figure 4 b). However, DPC-48 could not reproduce the intensity of convection similar to CPC-48 and the magnitudes of predicted CAPE values were relatively low in the former case (<1200 J/kg, Figure 4c). In this case, downscaling of grid resolution did not help in any improvements towards inference of severe weather event 48 hours in advance as the forecast fields of CAPE did not

indicate presence of deep convection over the Arabian sea, which was actually not true for $03^{rd}$ December 2017. DNC-48 yields very intense convective activities (CAPE >2500 J/Kg) for a very widespread region (Figure 4 d), and predicted CAPE fields were similar to the CPC-48 simulations as both of them could capture fair probability of deep convection over the Arabian Sea.

Figure 4 (e), (f) and (g) depicts COSMO forecasts based on the initial conditions of 12 UTC of $01^{st}$ December 2017, exactly

36 hours before the convective event of 00 UTC of $03^{rd}$ December 2017. Though downscaled simulations (i.e., DPC-36) reproduce features very similar to that seen in CPC-36 and DNC-36, the large magnitudes of CAPE indicating deep convection was confined to a smaller region in the case of DPC-36, whereas the CPC-36 and DNC-36 predicted widespread convective activities. The above mentioned features remained almost similar for the simulations based on initial conditions of 00 UTC of $02^{nd}$ December 2017 (Figure 4 (h), (i) and (j)) and 12 UTC of $02^{nd}$ December 2017 (Figure 4 (k), (l) and (m)) with DNC

yielding larger values of CAPE compared to the other two set of simulations.

It is important to note the differences between DPC and DNC in simulating the magnitudes of CAPE. Compared to DPC, DNC simulations could capture the fine scale features such as the spiral arms as well as the updrafts and downdrafts over the cyclone eye. DNC simulations always yield wide-spread coverage of deep convection compared to the DPC, even with a lead time of 48 hours. Figure 5 depicts model simulated areal averaged CAPE valid for 00 UTC of $03^{rd}$ December 2017 based on

 

different initial conditions for all the three numerical experiments. Since OCKHI was not stationary, and it was consistently progressing within the study domain, we have taken areal average of CAPE to represent the degree of convection prevailing over the region. It is a fair as well as a genuine approach to depict the convective event over widespread oceanic region because the core of the cyclone was travelling north-westward and the associated hot-spot of convection was also moving. As per the

spatial maps of CAPE obtained from ERA interim fields corresponding to 00 UTC of $03^{rd}$ December 2017, intense convective activities were prevailing for a widespread region with a peak value of CAPE exceeding 2500 J/Kg. Areal averaged CAPE for the model domain was about 500 J/kg (Figure 5). Similar features of atmospheric convection were seen in the CPC and DNC simulations though the areal averaged magnitudes of CAPE were somewhat lower than that seen in ECMWF Reanalysis fields. As discussed in previous paragraph, DPC simulations did not indicate the presence of deep convection which is again evident

from the low values of areal averaged CAPE seen for DPC simulations as against moderate values of CAPE resulted in CPC and DNC simulations. As described in the previous paragraph, areal averaged CAPE obtained from CPC and DNC simulations are found to be closer to the actual observations indicated by ECMWF fields. It may also be noted that the CAPE magnitudes obtained from ECMWF fields were always overestimated and the reasons are discussed in the next section.

### 5.1.2 Simulation of Precipitation Fields for 00 UTC of $03^{rd}$ December 2017

Since the identification of a deep depression over the Comorin Sea and its subsequent intensification to a CS and ultimately a VSCS, the coastal Arabian Sea as well as the southern Indian Peninsula experienced heavy to very heavy precipitation between $01^{st}$ to $03^{rd}$ December 2017. As the OCKHI moved north-westward on $03^{rd}$ and $04^{th}$ December, the intensity of rainfall got reduced over the coastal belt but it got increased over the oceanic region. In general, 24h accumulated rainfall extracted from the ECMWF Reanalysis fields were reasonably overestimated in comparison to the actual observations reported by IMD.

For instance, the Reanalysis fields corresponding to $05^{th}$ December 2017 indicated widespread rainfall over the Arabian Sea as well as the Kerala land mass, with intensity ranging from 0.1 mm to 15.5 mm (very light rain to light rain). However the actual observations reported by IMD indicated isolated light rains for a very small region within Kerala and a majority of the remaining landmass was rain-free and dry (Subrahamanyam et al., 2018). Most of the severe convective events associated with a progressing cyclonic storm remain dynamic and the underlying region with maximum influence also keep changing. In this

regard, the coarse grid ECMWF reanalysis fields of CAPE and precipitation have a tendency to express a widespread area with deep convection, whereas the actual region of precipitating zone may be very small compared to the grid resolution of global model itself. This leads to an overestimation of CAPE and precipitation in the ECMWF reanalysis fields while the regional model COSMO, which is configured for a relatively finer grid-resolution, attempts to constrain the area of deep convection to a smaller mesoscale region only.

Figure 6 (a) shows 24 hours accumulated rainfall between 00 UTC of $02^{nd}$ December 2017 and 00 UTC of $03^{rd}$ December 2017, inferred from the ECMWF Reanalysis fields. The intense precipitation zone associated with the progression of OCKHI led to heavy to very heavy (64.5 mm to 115.5 mm, and >115.6 mm) accumulated rain over the study domain on $03^{rd}$ December 2017. Even though observed precipitation fields for the land mass were reasonably low compared to the ECMWF Reanalysis fields, intense convective activities were prevailing over the study domain. On one hand, the CPC and DPC simulations indicate





widespread precipitation over the study domain, whereas the spatial maps of accumulated rainfall were confined to a minimal region for DNC. Even if the degree of convection over the study domain was predicted to be intense, but all the grid points do not yield sufficient proxy towards accumulated rainfall.

Figure 7 is an outcome of Figure 6, wherein spatially averaged 24h accumulated rainfall for different set of simulations,
together with the ECMWF reanalysis fields are depicted as histograms. From Figures 7 and 6, it can be seen that both the CPC and DPC simulations provide very similar pattern as well as quantity of the precipitation over the study domain, whereas the DNC simulations yield reasonably low magnitudes of precipitation. It may also be noted that the actual amount of precipitation simulated by COSMO is not only a function of Tiedtke convection parameterization scheme alone but also depends on other prognostic variables. With regard to the lead time requirements for prediction of intense rainfall associated with the passage of
cyclonic storm, both the CPC and DPC do not yield any significant differences except for the fact that high amount of rainfall is reproduced for the least lead time (i.e., +24 hours). The accumulated rainfall with a lead time forecast of +36 hours and +48 hours were marginally lower than that predicted by +24 hours (Figure 6 and 7). In the case of DNC simulations, even though the atmospheric conditions in terms of CAPE were more conducive for deep convection, it did not yield rainfall for all the grid points, and the area of precipitation zone was confined to the eye of the cyclone and closely adjoining regions within 500 kms.
From a comparison of CAPE and precipitation fields simulated from three distinct numerical simulations, it is clear that large values of CAPE reveal proximity of a deep convective episode but it is not necessary and essential that the available moisture in the lower atmosphere will yield heavy to very heavy precipitation. Downscaling of grid resolution helped in direct treatment of convection which was clear from the DNC simulations that reproduced high values of CAPE in the absence of a parametric approach. However switching off the scheme of convection parameterization did not help in reproducing intense rainfall over
the Arabian Sea.

Subrahamanyam et al. (2018) have shown a mean deviation of about 74 km between the predicted and observed eye of the cyclone for a lead time of 24 h from the genesis of OCKHI to its final dissipation stage. Here, we have evaluated the deviations between the predicted and observed eye of the cyclone for different initial conditions. Table A2 summarises the impact of explicit treatment of moist convection on the accuracy of predictions in terms of identification of the eye of the cyclone. Even
though the prediction of low pressure regime associated with the eye of the cyclone is not directly related with the treatment of moist convection, we notice a considerable improvements for the DNC simulations, which yield an error of about 37 km for a lead time of 24 h against an error of 78 km for the DPC simulations. Here, the DPC and CPC simulations resulted in very similar deviations, hence CPC simulations are not listed separately. While, the detection of eye of the cyclone is certainly improved for a lead time of 24 h and below, we do not see any reasonable improvements in the deviations for 36 h and 48 h
lead time. Large deviations ranging from 160 km to about 216 km for the COSMO model predictions with a lead time of 36 h to 48 h is attributed to inadequate model domain size which does not account for appropriate interactions between the model domain and the lateral boundaries. For improvements in the predictability of eye of the cyclone for a lead time of more than 24 h, the model domain must also be extended so as to account adequate impact of lateral boundary conditions on the model simulations. However, due to our computatinal infrastructure limiations, we have restricted our simulations to a moderate size
spatial region as shown in Figure 2.



## 6   Conclusion

In this research work, we assess the value of dynamical downscaling by examining the impact of convection parameterization scheme using COSMO in the presence of a VSCS over the coastal Arabian Sea. We have also attempted to investigate the role of initial conditions in prediction of severe weather alerts by providing different initial conditions, which helped us in

determination of the intensity of convection and its spatial spread across the study domain. This aspect becomes one of the most debatable topics in the area of parametrization, as there is no acceptable clarity about the spatial grid resolution of an atmospheric model at which the convective processes can be resolved directly. There is a good consensus that the use of high horizontal resolution improves weather and climate simulations in many ways, and tropical cyclones can be better resolved with finer grid spacing. In this regard, dynamical downscaling of NWP model's grid resolution by providing fine features of

surface-layer parameters such as roughness lengths, vegetation, soil types, etc. provides a more accurate topographical bound-ary conditions with increased resolution, particularly when they tend to resolve convective processes directly, rather than a parametric approach. Fine representation of orographic representation leads to improved precipitation pattern, but the present study over the oceanic region yields deterioration in precipitation forecasts in terms of accumulated rainfall magnitudes. How-ever, dynamical downscaling by switching off the convection scheme led to improved prediction of the cyclonic event in terms

of CAPE, which was able to provide severe weather alerts almost 48 h before the episode. Whereas a parametric treatment of convection with downscaling could not indicate presence of deep convective event on $03^{rd}$ December 2017 over the coastal Arabian Sea.

In one of the regime-dependent evaluation study of the accumulated precipitation in COSMO, Akkermans et al. (2012) have evaluated two variants of the COSMO: a fine-resolution version (2.8 km, COSMO-DE) and a coarse-resolution version (7

km, COSMO-EU) for the German region. They observed a strong over-estimation of the precipitation fields by COSMO-EU, whereas a positive bias was identified on top of topographical features (i.e., mountains or hill ridges). One of the plausible causes for discrepancies in the convection parameterization scheme was discussed as the graupel scheme in COSMO-DE, which might lead to over-estimation of the precipitation subject to formation of quick precipitation in the form of graupel (Akkermans et al., 2012). In this context, our investigation yields reduced precipitation over the coastal Arabian sea when

the convective processes are dealt explicitly and parametrization scheme for moist convection is switched off. Champion and Hodges (2014) have investigated some of the important problems associated with dynamical downscaling of the Met Office Unified Model by looking at the optimum configuration for obtaining the distribution and intensity of a precipitation field to match observations. Their study has shown that realistic precipitation intensities can be obtained using a high-resolution Lim-ited Area Model driven from a coarse resolution global model.

There is a visible increase in the frequency of severe or extreme weather events, such as cyclonic storms over the tropical oceans, and ability of a weather prediction model to predict these extremes is crucial for public safety and also to study the impact assessments caused by these events. In NWP models, dynamical downscaling will be a useful approach to capture and asses the potential impact of these severe weather events. It is an expensive affair to run NWP models with a finer resolution than necessary. The doubling in horizontal resolution and an accompanied reduction of the model's time step will lead to an





increase in computational demands by a factor of 8. Hence, it is important to understand under what conditions the dynamical downscaling is really capable of improving the simulations or can reproduce weather extremes in a satisfactory manner compared to the coarse grid model. Several studies have shown that downscaling improves the accuracy as we expect. But there are also examples about the cases where little improvements or sometimes deterioration of the forecast products after dynamical

5    downscaling. It may be because the benefits of high-resolution model seem to be highly case or geographically dependent and in some case due to issues with parametrization schemes. Most of the physical parametrization schemes in the NWP models are tuned or more suitable to a particular resolution or range of scales. In this context, the present article provides a valuable impact analysis of dynamical downscaling of COSMO model on the treatment of moist convection.





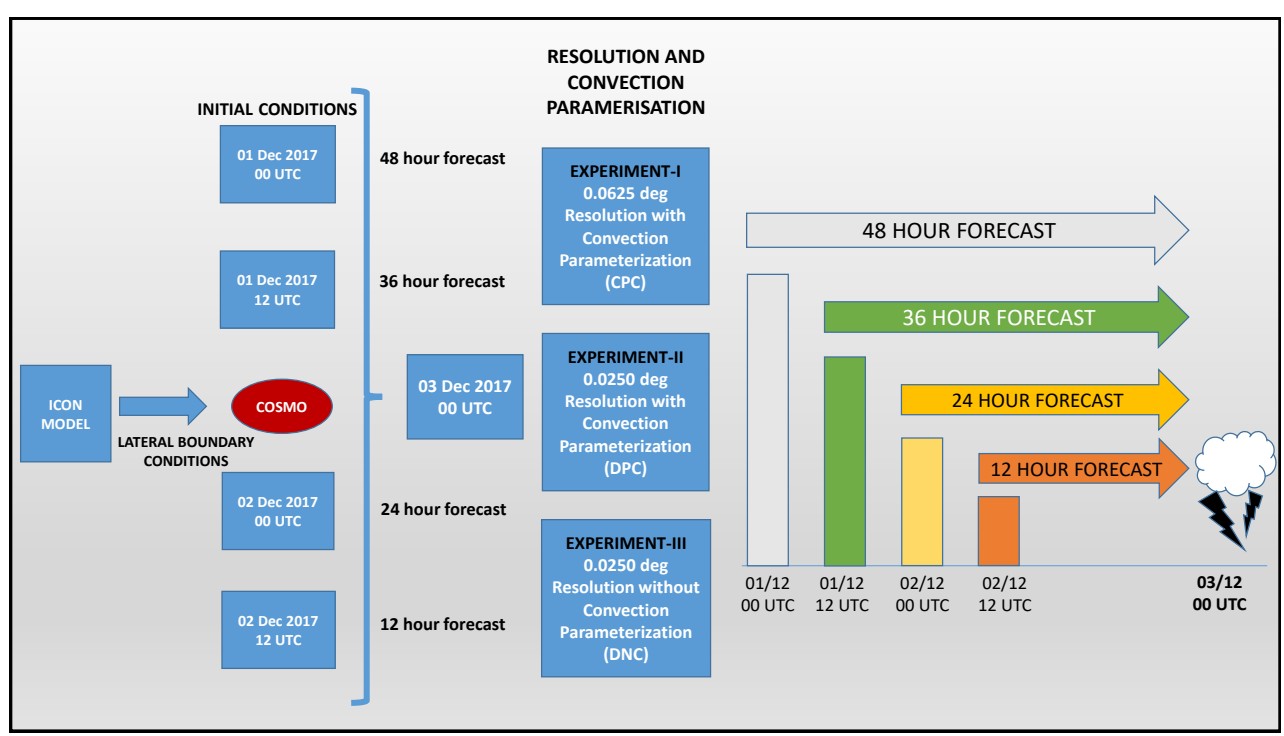

**Figure 1.** Schematic representation of three unique numerical experiments namely: (1) CPC (Controlled Parameterized Convection); (2) DPC (Downscaled Parameterized Convection); and (3) DPC (Downscaled with No parameterization of Convection). Ultimately all the three numerical experiments are targeted to simulate intense convective episode and associated precipitation for 00 UTC of $03^{rd}$ December 2017.



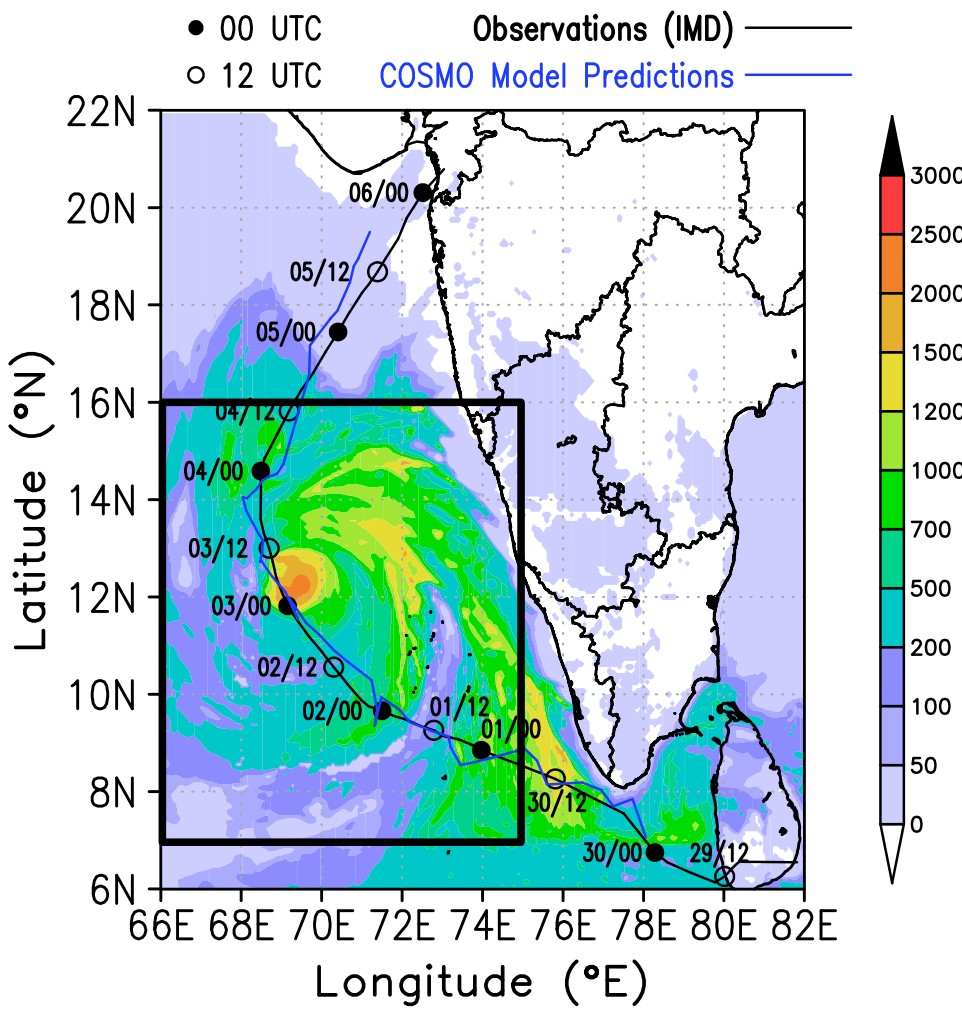

**Figure 2.** Geographical domain of the COSMO model (indicated within the black box). The background colour maps indicate CAPE (J/Kg) magnitudes simulated by COSMO for 00 UTC of $03^{rd}$ December 2017. The trajectory of OCKHI cyclonic storm from its formation as a depression on 29th November 2017 to its landfall and dissipation on $06^{th}$ December 2017 is also overlaid on the plot.

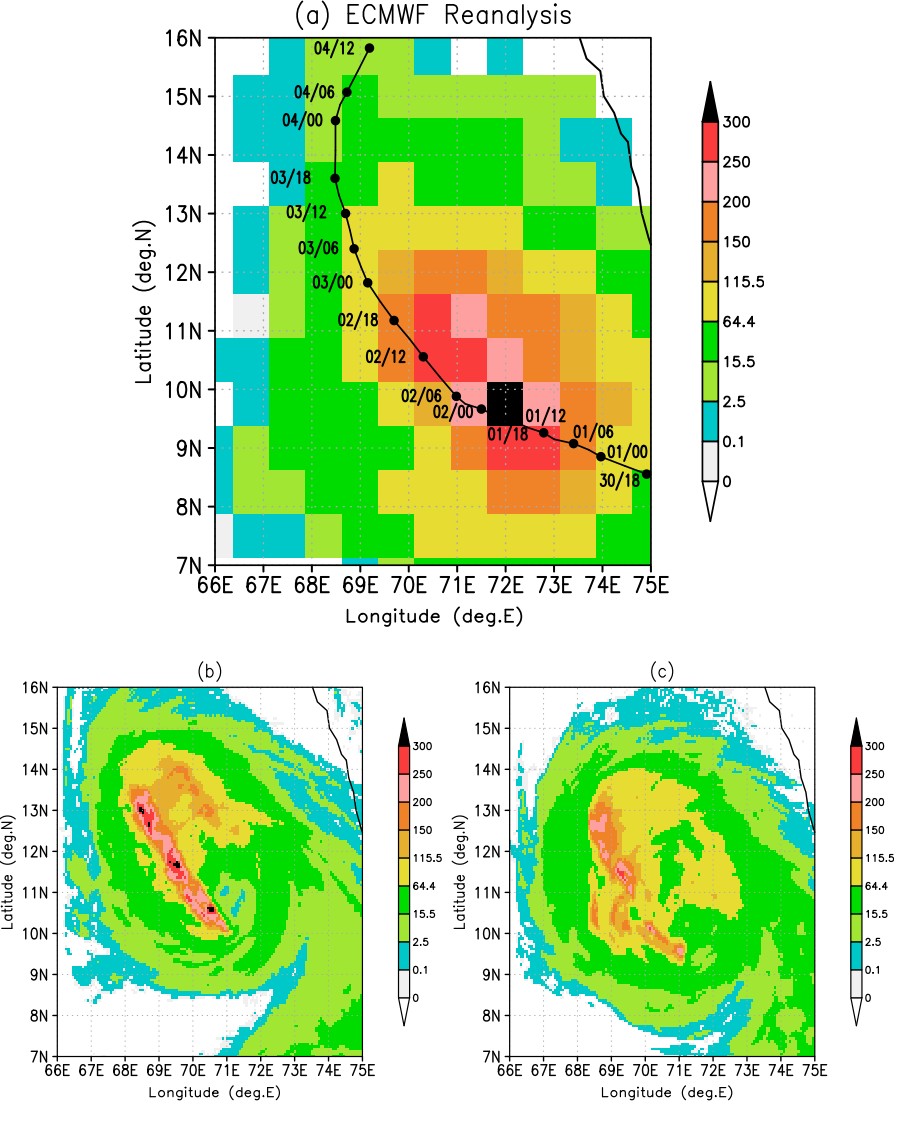

**Figure 3.** (a) ECMWF-ERA Reanalysis fields of 24 h accumulated precipitation (in mm) received over the study domain between 00 UTC of $02^{nd}$ December 2017 to $03^{rd}$ December 2017 associated with the passage of OCKHI. (b) +48 hours advanced COSMO forecasts of concurrent precipitation field based on the initial conditions of 00 UTC of $01^{st}$ December 2017. (c) +24 hours advanced COSMO forecasts of concurrent precipitation field based on the initial conditions of 00 UTC of $02^{nd}$ December 2017.



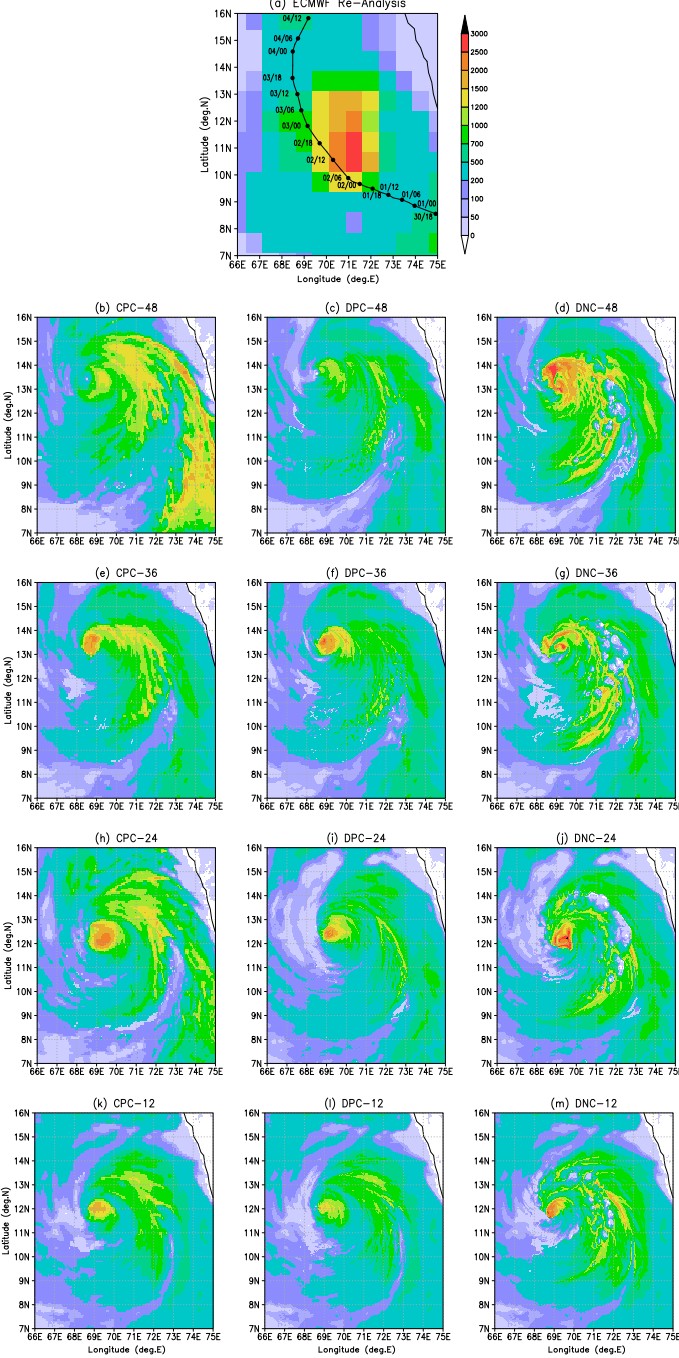

**Figure 4.** (a) CAPE (in J/Kg), one of the stability indices for inferring the degree of convection, valid for 00 UTC of $03^{Rd}$ December 2017 extracted from the ECMWF Reanalysis fields; (b), (c), (d) second panel from the top - Concurrent +48 hours predictions of CAPE based on CPC, DPC and DNC simulations respectively. Third, fourth and fifth panel from the top- same as second panel, but for +36, +24, and +12 hours respectively.

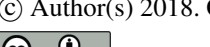



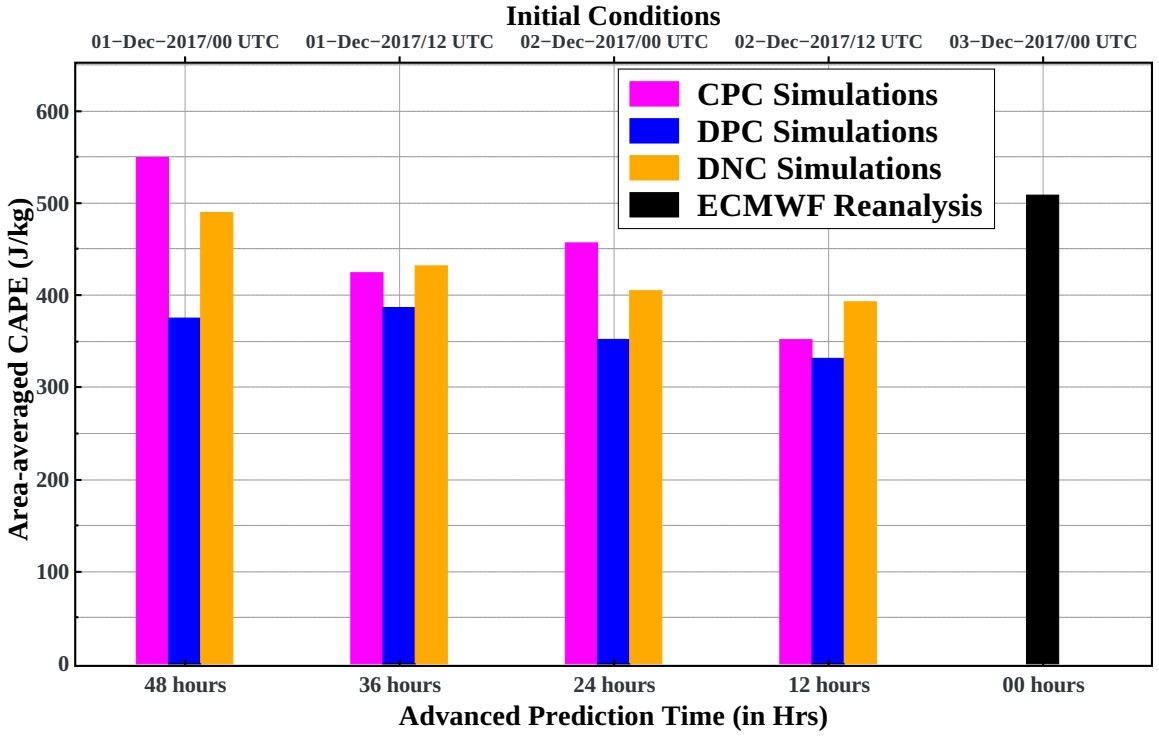

**Figure 5.** Area averaged CAPE (in J/Kg) from three set of simulations; CPC, DPC and DNC, valid for 00 UTC of $03^{rd}$ December 2017.
Date and Time of the initial conditions used for these experiments are marked on the top axis. Actual magnitudes of CAPE extracted from
ECMWF Reanalysis are also depicted in the plot



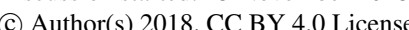

**Figure 6.** (a) 24 hours accumulated Rainfall (in mm), valid for 00 UTC of $03^{Rd}$ December 2017 extracted from the ECMWF Reanalysis fields; (b), (c), (d) second panel from the top - concurrent +48 hours predictions based on CPC, DPC and DNC simulations respectively. Third and fourth panel from the top- same as second panel, but for +36 and +24 hours respectively.





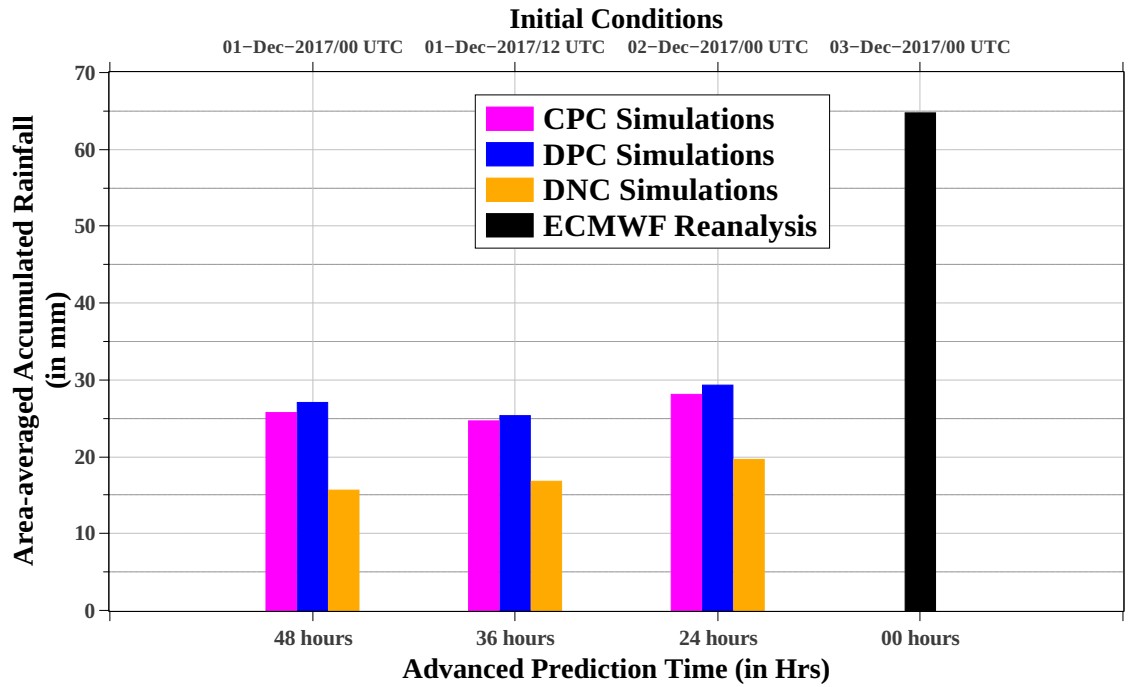

**Figure 7.** Area averaged 24 hour accumulated rainfall (in mm) from three set of simulations; CPC, DPC and DNC, valid for 00 UTC of $03^{rd}$ December 2017. Date and Time of the initial conditions used for these experiments are marked on the top axis. Actual magnitudes of CAPE ext acted ECMWF Reanalysis are also depicted in the plot



**Table A1.** Technical Description of the COSMO Model used in the present study

| TECHNICAL DESCRIPTION OF THE COSMO MODEL USED IN THE PRESENT STUDY | | | |
|---|---|---|---|
| **MODEL DOMAIN AND REFERENCE ATMOSPHERE** | | | |
| Study domain | Coastal Arabian sea | | |
| Longitudinal Coverage | 66°E to 75°E ($\sim$ 1000 kms) | | |
| Latitudinal Coverage | 7°N to 16 °N ($\sim$ 1000 kms) | | |
| Reference atmosphere | Default version of the COSMO | | |
| **PARAMETERS FOR THE MODEL RUN** | | | |
| Initial conditions of atmosphere | ICON (German global model) analysis fields | | |
| Lateral boundary conditions | Time-varying ICON forecast fields at +3 h regular intervals | | |
| Duration of forecast | +48 h with hourly dumps of forecast fields | | |
| **PARAMETERS FOR THE DIABATIC MODEL** | | | |
| Grid-scale precipitation | Kessler-type warm rain parameterization scheme with cloud water and cloud ice | | |
| Radiation | Ritter-Geleyn radiation parameterization scheme Ritter and Geleyn (1992) | | |
| Vertical turbulence diffusion | Prognostic TKE-based scheme, which includes effects from sub grid-scale condensation/evaporation | | |
| Surface-layer turbulent fluxes | New TKE-based scheme including a laminar sub-layer | | |
| Soil and vegetation processes | Multi-layer version based on Jacobsen and Heise (1982) | | |
| **PARAMETERS FOR THE ADIABATIC MODEL (NUMERICS)** | | | |
| Time-integration scheme | Two time-level Runge-Kutta scheme with time-split treatment of acoustic and gravity waves | | |
| Horizontal diffusion | 4th-order linear scheme with orographic limiter | | |
| Treatment of lateral boundary conditions | Free-slip lateral boundary conditions for $w$ in case of non-periodic Davies-type lateral boundaries | | |
| **TYPE OF NUMERICAL EXPERIMENTS** | | | |
| | **CPC** | **DPC** | **DNC** |
| Horizontal Grid Resolution (Along Longitudes) | 0.0625°($\sim$ 7.0 km) | 0.025°($\sim$ 3.0 km) | 0.025°($\sim$ 3.0 km) |
| Horizontal Grid Resolution (Along Latitudes) | 0.0625°($\sim$ 7.0 km) | 0.025°($\sim$ 3.0 km) | 0.025°($\sim$ 3.0 km) |
| Number of Grid-points along Longitudes | 256 | 361 | 361 |
| Number of Grid-points along Latitudes | 256 | 361 | 361 |
| Number of Grid-points in vertical along Altitudes | 50 | 50 | 50 |
| Model time step (in seconds) | 60 | 30 | 30 |
| **DATE AND TIME OF INITIAL CONDITIONS, AND FORECAST DURATION** | | | |
| 01-December-2017: 00 UTC, 48 h | CPC-48 | DPC-48 | DNC-48 |
| 01-December-2017: 12 UTC, 36 h | CPC-36 | DPC-36 | DNC-36 |
| 02-December-2017: 00 UTC, 24 h | CPC-24 | DPC-24 | DNC-24 |
| 02-December-2017: 12 UTC, 12 h | CPC-12 | DPC-12 | DNC-12 |



**Table A2.** Differences (in km) between the IMD reported coordinates of the eye of the cyclone and that simulated by DPC and DNC

| Initial Conditions | Difference in Cyclone eye coordinates between IMD and COSMO DPC run (in km) | Difference in Cyclone eye coordinates between IMD and COSMO DNC run (in km) |
|---|---|---|
| 48 hour forecast | 216.323 | 190.104 |
| 36 hour forecast | 201.223 | 160.924 |
| 24 hour forecast | 77.993 | 36.550 |
| 12 hour forecast | 31.230 | 30.024 |





*Author contributions.* The design of three distinct simulations were conceived by RS and were subsequently modified by DBS and TJA. The numerical experiments with regard to COSMO were conducted by RS under the supervision of DBS and RR. The necessary observational data from ECMWF were also analysed by DBS and RS. The first draft of manuscript was prepared by RS and was duly corrected by DBS, RR and TJA.

*Competing interests.* The authors declare that they have no conflict of interest.

*Acknowledgements.* The COSMO model is freely available to the users community for scientific and research purpose, and the model details can be accessed through its website (http://www.cosmo-model.org/). Space Physics Laboratory (SPL) has a scientific license for utilisation of the COSMO model in research mode, and the authors are thankful to the Deutscher Wetterdienst (DWD, German Weather Service) for providing the initial and lateral boundary conditions from the ICON global model for this study. We express our sincere gratitude to Drs.
Ulrich Schattler, Detlev Majewski and their colleagues from Deutscher Wetterdienst, Germany for their continuous support in setting up of COSMO at SPL. The ERA-Interim Reanalysis data fields used in this article are part of ECMWF's Meteorological Archive and Retrieval System (MARS), which is accessible to registered users in ECMWF Member States and Cooperating States from the ECMWF Data Server at http://data.ecmwf.int/data. Authors duly acknowledge the ECMWF for making the reanalysis fields available in the public domain through their services. One of the authors Roshny is thankful to the Indian Space Research Organisation for her PhD research fellowship. Anurose T
J was supported by the Federal Ministry of Education and Research in Germany Bundesministerium fuer Bildung und Forschung; BMBF) through the research program High Definition Clouds and Precipitation for Climate Prediction - HD(CP)2.




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
