# Peer review of "Impact analysis of dynamical downscaling on the treatment of convection in a regional NWP model - COSMO: a case study during the passage of a very severe cyclonic storm "OCKHI""

_Natural Hazards and Earth System Sciences, 2018_

## Referee Comment (RC1) · Anonymous Referee #1 · 19 Jul 2019

Review of "nhess-2018-288"

**Title: Impact analysis of dynamical downscaling on the treatment of convection in a regional NWP model – COSMO: a case study during the passage of a very severe cyclonic storm "OCKHI"**

**Authors: Roshny S. D. Bala Subrahamanyam, T. J. Anurose, and R. Ramachandran**

The study evaluates the representation of a cyclone over the Arabian Sea in COSMO model simulations at different horizontal resolutions and different treatments of convection. More specifically, the authors performed simulations at a grid spacing of (i) 0.0625° with a parameterized convection, (ii) 0.025° also with a convection scheme, and (iii) 0.025° with explicit convection. Precipitation and CAPE fields from COSMO are then compared to ERA-Interim reanalysis data in an attempt to evaluate which model configuration better represents the convection and precipitation during the passage of the cyclone over the Arabian Sea along the Indian Peninsula.

Overall, I found the study potentially relevant and the manuscript carefully written. However, there are several major flaws in the model setup of the experiments, the meteorological evaluation, and the choice of the data set that serves for the model evaluation. Therefore, a substantial part of the analysis is invalid and the conclusions remain unsubstantiated. In my view, the required corrections go beyond major revisions. However, the study has much potential when these major comments are accommodated. Tropical cyclones cause frequently severe socioeconomic impacts, and their simulation and predictability are of high interest to the scientific community and operational weather forecasting. Also, the model experiments with different representations of convection are relevant and deserve attention. Therefore, I would like to encourage the authors to implement these changes and to resubmit the manuscript.

**Major comments**

**1. Dynamical downscaling**

At several occasions, including the title, the abstract and the conclusions, the manuscript claims to assess the impact of dynamical downscaling on the representation of convection in the numerical weather prediction model COSMO. Dynamical downscaling refers to the use of a limited area model or regional climate model to provide more detailed information on weather or climate that cannot be provided by a global weather or climate model that typically has a relatively coarse model resolution. When claiming that the impact of dynamical downscaling is assessed, the study would need to show a comparison between the global data (from ICON) and the limited area model (COSMO), for example, for the simulation of precipitation and CAPE. The present manuscript does not show any data from the global model. In order to evaluate impacts of dynamical downscaling, as the manuscript claims to address, the authors would need to include a comparison between the ICON and COSMO data, and then to use observation-based data to show that the dynamical downscaling indeed leads to an improvement.

At other places in the manuscript, for example in Section 4, the text seems to imply that the use of different horizontal grid spacing aids the evaluation of the impacts of dynamical downscaling. This is incorrect as it can only help to investigate the sensitivity of convection to the model resolution. In other words, the use of different horizontal model resolutions is not the same as dynamics downscaling.

**2 Observations for model validation and comparison**

The study uses the ERA-Interim data set from the ECMWF as a means to validate the COSMO model simulations. This approach is problematic since the ERA-Interim reanalysis is produced at a resolution of about 0.7°x0.7°, much coarser than the COSMO model simulations which use a horizontal grid space of about 3-7 km. Later, in the analysis it becomes indeed clear that the data is much coarser (e.g., page 8, lines 26-27) and that the center of the cyclone is more off in the ERA-Interim data than in the COSMO simulations. As a consequence, the comparison between the COSMO simulation experiments and ERA-Interim as shown in Figures 5 and 7 is not relevant. Therefore, the conclusions based on this comparison, as for example phrased in the last sentence of the abstract, are not supported by a valid analysis.

I highly recommend to use precipitation observations based on satellite estimates, for example TRMM (Huffmann et al., 2007) or any other satellite product, whereas CAPE fields from the operational IFS analysis from ECMWF may provide higher resolution data than the ERA-Interim reanalysis.

**3. Model domain**

The used model domain is very small with only 10 degrees / 1000 km in zonal and meridional direction. In fact, the cyclone is located near the boundary of the domain at the initial time of the simulations with a +48 and + 36 hour lead time, as shown in Figure 2. This model configuration is problematic for obtaining proper results. I recommend using a model domain that is sufficiently covers the tropical cyclone throughout the simulation. As also written in section 5.1.2 (page 11 and lines 30-32), it is understood that computational resources can be a limitation; however, this cannot justify a model simulation that does not support a valid study.

**4 Simulation experiments**

The study uses three different simulation experiments; (1) with a grid spacing of 0.0625º (~7 km) and convection parameterized, (2) with a grid spacing of 0.025º (~3 km) and convection parameterized, and (3) with a grid spacing of 0.025º (~3 km) and without convection scheme. Following previous studies, convection schemes can potentially be switched of at the order of a 7 km grid spacing, whereas convection may largely be resolved when using a grid spacing of 3 km (e.g., Marsham et al., 2013). This is also explicitly stated in the manuscript on page 7, lines 17-18. The results show that experiment (2) does not add much to experiment (1), whereas the convection-permitting simulation of experiment (3) shows a lot of details as compared to experiment (2). Therefore, I would recommend to replace experiment (2) by a simulation with a grid space of 0.0625º (~7 km) and without the use of a convection scheme.

**5. Meteorological analysis.**

The analysis in the manuscript is limited to CAPE and precipitation. In order to learn more about the representation of the tropical cyclone in the different model simulations, it may be helpful to extend the analysis with other meteorological variables, for example, sea level pressure and equivalent potential temperature. In particular, SLP can reveal information about the track of the cyclone, which can be compared to the observations and thus be a measure for the accuracy of the different model simulations.

In addition, the area averaged amounts of precipitation and CAPE, as shown in Figures 5 and 7, may not be appropriate for a comparison of the model simulations to observational data. The area averaging leads to a loss of detailed information and can be misleading as a comparison.

**Minor comments**

At several places (e.g., lines 2, 5 and 8 on page 3), the text speaks about a cyclone. Is there a specific reason why not to speak about a tropical cyclone? The term "cyclone" is a very general term that also covers extratropical cyclones that are found in the extratropics.

Page 1, lines 19-20. Instead of speaking about "the smallest and most compact weather processes", please, speak in terms of spatial and time scales.

Please, omit lines 17-18 on page 2; "With the availability … becoming finer.". The use of higher model resolutions is primarily limited by computational resources, not by the availability of observations.

At several places in the manuscript (e.g., page 3, lines 10-11, page 8, line 9, and page 12, lines 3-4) the text speaks about simulation experiments with different initial conditions. This can be understood by readers as use of slightly perturbed initial conditions at the same date, as for example used for ensemble simulations. Instead, the difference between the simulations are different forecast lead times with respect to the episode of interest. Please, rephrase the text where needed.

Section 2. Please, specify which COSMO version is used.

Page 4, lines 12-13 The phrase "Since the convective processes … … much smaller than those resolved by mesoscale and regional NWP models" is not entirely correct. In case of high-resolution simulations, for example, with a horizontal grid spacing of 3 km, convective processes may be largely resolved by the model.

Page 5, lines 8 "to the standard version of the model"; what is the standard version of the model in their study? Please, clarify in the text.

Section 3.1 Please, state in the text how many model levels are used for the model simulations.

Page 5, line 28 speaks about the use of ICON global model data for a period of 9 days. This is not consistent with the ICON model data that are used for four different time instances on 2 days as mentioned later in the text (Page 6, line 10). Please, correct.

Page 6, lines 13-18. Please, omit the technical description "COSMO model … … fine-grid resolution."

Page 7, lines 10-12. Why don't you show the precipitation from ICON? See also major comment number 1.

Page 7, line 26. In what way was the tropical cyclone rare? In its intensity, duration and / or socioeconomic impacts? Please, be specific.

Page 8, lines 12-13. Please, state the source of these precipitation observations.

I recommend to restructure sections 5.1.1 as 5.2 and 5.1.2 as 5.3.

Page 9, line 4. It is not only the state of the lower troposphere that defines CAPE. Please, replace "lower atmosphere" by "the thermodynamic conditions".

Page 9, lines 6-7. Please, clarify which forecast step is used from ERA-Interim.

Page 9, lines 13-14 "The ECMWF fields were almost off by more than 100 kms from ..."and page 10, lines 12-13 "… the CAPE magnitudes obtained from ECWMF fields were always overestimated … " and page 10, lines 24-29 "In this regard …  … a smaller mesoscale region only". This shows that the ERA-Interim data is not suitable for validation of the model simulations, see also major comment number 2.

Page 10, line 32. Are these precipitation amounts per day?

Page 11, lines 2-3. I cannot follow the sentence "Even if … accumulated rainfall". Please, rephrase.

The conclusions at page 11, lines 19-20 "However, switching off … … rainfall over the Arabian Sea." and at page 12, lines 12-13 "Fine representation … … accumulated rainfall magnitudes." are invalid due to the comparison of COSMO simulations to ERA-Interim data. Satellite-based estimates may provide a base for a more realistic and useful comparison, see also major comment number 2. Moreover, Figure 6 shows that the DNC simulation has intense rainfall, although area-averaged amounts as in Figure 7 may be lower as compared to ERA-Interim.

Page 12, lines 30-31. The sentence "There is a visible increase … … over the tropical oceans" needs to be supported by references or otherwise be removed.

**Writing comments**

Page 1, line 5. Please, consider to replace "inter-linking of" by "interplay between".

Page 1, line 15. Please, replace "an NWP" by "a NWP" and write out NWP. Abbreviations used in the abstract need again to be defined within the text of the manuscript upon first use.

Page 1, line 20 – Page 2, line 1. Please, replace "surface to the troposphere" by "surface to the upper troposphere"

Page 2, line 3. Please, replace "Convection process" by "Convective processes" or "Convection".

Page 2, lines 9-10. Please, rephrase the sentence "However, the process of convection … interaction with radiation", for example, as "Moreover, convection involves complex interactions with cloud formation which influence the atmospheric circulation through radiative effects." or in a similar direction.

Page 2, line 12. Please, replace "inter-linked" by "complex".

Page 2, line 13. Please, replace "Further" by "Furthermore, ".

Page 2, line 16. Please, replace "is apparently inter-linked with" by "constrained by" or in a similar direction.

Page 2, line 19. Please, replace "As on today" by "At present" or "Currently".

Page 2, line 26. Please, remove "physical and".

Page 3, lines 16-17. Please, correct the sentence by writing "COSMO (formerly known as … …in Switzerland) is a non-hydrostatic … … model that was initially ...".

Page 4, line 9. Please, replace "the conserved framework" by "the conservation of" and in line 10, replace "Different schemes" by "Schemes".

Page 4, line 14. Please, replace "resolved" by "estimated" or "represented".

Page 4, line 18. Please, replace "cumulus" by "convective".

Page 4, line 21. Please, replace "the Convectively" by "Convective".

Page 4, line 25. Please, replace "COSMO model" by "The COSMO model".

Page 4, line 30. Please, replace "lies" by "lie".

Page 5, line 12. Please, remove the comma after "that".

Page 5, line 11. Please, replace "can provide improved" by "can improve".

Page 5, line 31. Please, replace "form" by "from".

Page 6, line 11. Please, remove "and meteorological".

Page 6, line 27 as well as on page 7, line 8. Please, replace "under the framework of" by "using the" or "with the".

Page 7, line 1, 10, and elsewhere in the manuscripts. Always, use a comma, before "respectively".

Page 7, lines 2-3. Please, remove "to the COSMO model" and write "of the actual episode".

Page 7, lines 6. Please, remove "to the COSMO model".

Page 7, lines 15-17. Please, rewrite this long sentence. Stating that you switched off the convection scheme or use a convection-permitting simulation is sufficient.

Page 7, line 14. Please, rephrase "are treated directly" by "are explicitly simulated", or "permitted" or in that direction.

Page 7, lines 26 and 31. Please, rephrase "residence time" by "life span" of "life cycle".

Page 7, line 30. Replace "between" by "from".

Page 7, line 30 and at many other places in the manuscript. Rewrite "01st December" and "5th December" as "1 December" and "5 December".

Page 8, line 3. Please, replace "The CAPE" by "CAPE".

Page 8, line 7. Please, reverse "final dissipation and landfall".

Page 8, line 22. Please, replace "corresponding to" by "on".

Page 9, line 5. Please, replace "proactive" by "conducive" or "favorable".

Page 9, line 10. Please, remove ", and the category of storm was retained as VSCS.".

Page 9, line 18. Please, replace "downscaling of" by "a higher".

Page 9, line 20. Please, replace "which was actually not true for" by "occurred on".

Page 9, lines 24. Please, write "Figures … depict...".

Page 9, lines 27-28. Please, replace "was" by "were" and "activities" by "activity".

Page 10, line 15. Define the abbreviation "CS" or write full out.

Page 10, line 18. Remove "got" and "it got".

Page 10, line 19. Replace "reasonably" by "substantially".

Page 11, lines 9-11. Please, replace "requirements" by "experiments", "amount" by "amounts", "is" by "are", "least" by "shorter".

Page 12, lines 4-5. Please, replace "in determination of" by "to determine".

**Figures & Table**

Figure 1. Please, remove the words downscaling in the caption as all simulations are downscaled in the sense that the simulations are fed by global data. Instead, indicate the resolution of horizontal grid spacing of 7 and 3 km.

Figure 2. The caption speaks about CAPE from COSMO. CAPE fields extend outside the model domain that is indicated by the black box when I understand correctly. Is this perhaps CAPE from ERA-Interim?

Figure 6. Is this the accumulated rainfall in the 24 hours prior or after 00 UTC, 3 December 2017? Please, clarify.

Table A2. For the sake of consistency, I would recommend to also include the results from the CPC simulation.

**References**

Huffman GJ, Adler RF, Bolvin DT, Gu G, Nelkin EJ, Bowman KP, Hong Y, Stocker EF, Wolff DB. (2007), The TRMM multisatellite precipitation analysis (TMPA): Quasi-global, multiyear, combined-sensor precipitation estimates at fine scales. J. Hydrometeorol. 8: 38–55.

Marsham JH, Dixon NS, Garcia-Carreras L, Lister GMS, Parker DG, Knippertz P, and Birch CE. (2013), The role of moist convection in the West African monsoon system: Insights from continental-scale convection-permitting simulations. GRL, 40: 1843-1849.

---

## Referee Comment (RC2) · Ronny Petrik (Referee) · 1 Aug 2019

[a4paper,10pt]article [utf8]inputenc

[Figure]

**Review of the paper 'Impact analysis of dynamical ... a case study during the passage of a very severe cyclonic storm 'OCKHI'**

Ronny Petrik

August 1, 2019

**1   General comments**

The paper reviewed is about a very severe cyclonic storm in the Arabian Sea. In the framework of a downscaling experiment, the author investigates the impact of model resolution and convective parameterization on the results.

The English language in the text is IMO proper to achieve a good flow of reading and to get the context right. The structure is clear and the figures and tables are well done.

However, main issues appear with the text which call at least for a major review. If the author is not adequately tackling that issues, the scientific content will still be questionable (i.e. very likely a rejection of the content).

**1.1 The authors intention - analysis of sensitivity for initial conditions**

In the paper presented a sensitivity to lead times is done. However, to identify sensitivity for initial conditions, other forcing data have to be considered. In the case of the Arabian sea I would prefer ERA-analysis, ERA5-reanalysis, MERRA2 reanalysis or NCEP analysis as well as reanalysis. Thus, having three different types of analysis, the sensitivity study is much more convincing. One would incorporate the spread originating from the different physical parameterization schemes and the assimilation techniques.

**1.2 The authors intention - analysis of sensitivity for parameterization of convection**

From previous studies it is already clear that parameterization for deep convection can be switched-off for resolutions smaller than about 3-5 km. The interesting question is the 'about'. Therefore, I see no reason why to add the DPC experiment with 2.8 km resolution. However, the sensitivity analysis would get more meaningful if the author decides

- to deal with an experiment in the convective 'grey zone' and performs a simulation at 4 to 5 km with parameterization for convection switched-off and switched-on.

- to investigate the need for parameterization of shallow convection. That means to add a simulation at 2.8 km resolution deactivating the shallow convection (which is active in the standard configuration).

To clarify, the recent content of the paper is somehow '2.8 km resolution leads to more details in the CAPE and precipitation fields, compared to CPC experiment with 7 km resolution. The experiment DPC is unnecessary because the patterns are smoothed and the area-averaged precipitation is the same as for CPC. The CAPE values are
off compared to ERA-Reanalysis and DNC, CPC.' However, addressing the research questions the author mentioned in the introduction, it is required to go beyond the experiments introduced in the recent version of the paper.

**2 Evaluational basis**

The evaluation of the results is superficial. First, ERA-reanalysis data are not helpful in measuring the quality of the high-resolution model. The author should consider satellite data from TRMM as remote sensing observations. In addition, the data from IMD are referred but at no time a quantitative comparison is provided to the reader. Without such a comparison, the author cannot raise arguments like 'the downscaling did not improve rainfall prediction' or 'the CAPE magnitudes obtained from ECMWF fields were always overestimated'.

The basis for evaluation could be more improved by incorporating radiosonde data or satellite data about the cloud structures. The ERA-reanalysis can be useful for qualitatively analyzing those meteorological parameters, which are more or less instantaneously assimilated, as the mean sea level pressure.

**3 Robustness of the analysis**

The author confines himself to the analysis of precipitation and CAPE. Much more meteorological parameters have to be evaluated to get a clue about the differences in the model results and the related performances. It would be very beneficial to study the vertical structure of the cyclone along the path or as a cross section, to visualize the path of the eye (distance to observed position) for all configs in one figure, to look at the cloud structures, the simulated vertical velocity and the vertical integrated

cloud content as well as moisture-flux divergence (as a precursor for the convection parameterizations).

Furthermore, the idea of downscaling is to add some value to the forcing model, which is the ICON in your case. The author misses to analyze which of the configurations is superior over the forcing simulation. It is not fair and not useful to compare the high resolution simulations with a global reanalysis, which cannot hold as a reference for a 'global prediction' as well as an observational field. It is much too coarse compared to the models the author deals with.

In addition, I am asking myself why not to choose a model domain capable of resolving the initiation of the storm. I.e. the extension of the domain in Southern direction by 1 degree and in Eastern direction by 2 degrees captures the whole intensification stage of the storm. Doing so, one gets more independent from the global forcing regarding lateral conditions.

However, it is still a big challenge to extract some general scientific implications from a single case study for the scientific community. Thus, it would be worth to look at other comparable events which would extend the study in a reasonable manner and which would result in a more robust statistical and scientific basis.

**4   Specific comments**

**4.1   Introduction**

The introduction is well written and with a nice literature review. However, it is too general, i.e. a literature discussion about tropical storms is missing as well as the performance of models resolving them. Furthermore, I miss a section overview at the end.

page 1, line 17: 'to name a few' can be skipped

page 2, line 1: Start a new sentence

page 2, line 21: 'meteorological data' can be replace by data. The forcing data are much more than meteorological data (hydrological, ...)

page 2, line 31: Regarding the discussion of resolution needed to achieve a complete explicit representation of convection, the author should refer Bryan (2003) [Resolution Requirements for the Simulation of Deep Moist Convection].

page 2, line 34 - page 3, line 2: reading is lost due to large bracket text

**4.2 COSMO model**

IMO the section 'COSMO model' should be divided into '2.1. General description' and '2.2. Parameterization of Convection'

page 3, line 22: 'The equations are solved numerically on a Arakawa C-grid (Baldauf, 2011)' - this is all you need here. Everything else would be too complicated.

page 3, line 22: 'The temporal integration of the governing equation is done with' ...

page 3, line 23-24: Reformulate the sentence with the vertical layers. Please skip the number 50, because you are later on explaining the model configuration.

page 3, line 27-28: Please skip the sub-clause about diagnostic variables. This would be a list without end.

page 3, line 30-31: 'formation of precipitation fields' is a little bit too misleading. I would recommend to use 'The formation and modification of clouds and precipitating constituents'.

page 4, line 2-3: The sentence about Tiedtke can be skipped. The section 2.2. is

discussing all details about moisture convection.

page 4, line 14-18: The sentence is too long.

page 5, line 14-18: The first two sentences should be shifted to section 3. The last sentence should be placed in section 2.1 (I suggested).

**4.3 Methods and Data**

This section should be rearranged. At first, a renaming to 'Methods' and Data' would be beneficial. Second, a good naming of section 3.1. is IMO 'Configuration of the Model simulations'. Third, the recent Section 4 should be Section 3.2. named 'Sensitivity experiments with NWP model'. Fourth, the recent section 3.2. should be Section 3.3. 'Observations'. Regarding the COSMO model, it is needed to explicitly tell the version number. Having this version number, the community exactly knows about bugfixes and the state of research with your model version.

page 5, line 21: You do not explain 'VSCS'. I think it is very service convective storm.

page 5, line 26+27: Two commas would be helpful after 'km' and 'latitudes'.

page 5, line 30-31: I do not understand this last sentence here.

page 6, line 2: ERA data are not an observation. It is a model forced to the atmospheric state observed. This is fully different than an observation. You can call it a reanalysis. Not more like this.

page 6, line 10-19: This paragraph should be shifted to Section 3.1. 'Configuration of model simulations'.

page 7, line 2-3: Please skip everything starting from 'respectively'. You have already explained about that detail.

page 7, line 5-12: I never read before something complicated like this. Please reformulate that paragraph in such a way that it is clear 'only the resolution changes compared to CPC.

page 7, line 19-20: This last sub-clause is redundant information. You have already explained that for the other configurations.

4.4 Results and Discussion

IMO, this section consists of two subsections 4.1. and 4.2. The discussion about the location of the storm beginning at line 21 on page 11 is worth to put in an own subsection 4.3.

page 8: Where are the paragraphs here? One suggestion from my side: line 23.

page 8, line 20-23: This deviation of the path is fully misleading here. The location of the storm is discussed later and needs in my opinion an own section.

page 8, line 27: I cannot observe from Figure 3 that the magnitude of rainfall is larger for ERA, but the spatial extend of regions with a high amounts of rainfall is much larger in ERA than in COSMO. Furthermore, I see a shift in the maximum precipitation field between ERA and COSMO.

page 9, line 10: Is this an observation from a radiosonde? If so, it should be highlighted here because then the reader knows which value is realistic (and not only a model output).

page 9, line 12-13: You mention that the eye in COSMO forecast is 40 km away from observations. Yes, but the ERA is much far away from the observations. IMO, this discussion should be placed in 4.3. Otherwise, the information falls from the sky.

page 9, line 24-30: I miss the discussion about the placement of the CAPE maximum at Figure 4. It is evident that the runs with 24 lead times place the maximum more the South compared to the runs with longer lead times.

page 9, line 32-33: You argue something about downdrafts and updrafts, but no figure or detailed text is given. What do you exactly mean? What is a realistic downdraft and updraft? Such a discussion would be a chance to improve the paper and make it more scientific.

page 10, line 12: ERA is not observations. Please skip that.

page 10, line 12-13: You argue that the CAPE values of the ERA are always overestimated, but you do not give a proof for it. IMO, this sentence can be skipped.

page 10, line 15-23: There are no observation by the IMD shown. Thus, the reader has no feeling for the differences between model and observations.

page 10, line 27: the ERA reanalysis fields show an overestimation in spatial extend but without any observations the reader would not believe that magnitudes of precipitation and CAPE are overestimated.

page 11, line 2-3: I do not understand the meaning of that sentence.

page 11, line 5: This is a barplot and not a histogram.

page 11, line 7-9: A sentence without content. Please skip it.

page 11, line 9-12: The discussion about leadtime requirements is confined to precipitation intensities but not to location of intense precipitation. I do not understand, why this is less important. Regarding lead times, this is a crucial point.

page 11, line 19-20: Again, as already said, what is the value of such sentence without having seen any observation.

page 11, line 31-35: The critical discussion about predictability only includes the model domain. However, the quality of the initial and lateral boundary conditions is of much more importance, but it is not discussed and analyzed at all.

**4.5 Conclusions**

The conclusions are too general and off-topic. The main content is about preconditions for high-resolution simulations and improvements or problems detected in other studies. The relation to this paper is not so clear. IMO, the conclusion should be rewritten in order to get a clue about the implications of the author for the whole scientific community.

page 12, line 9-12: Too long sentence.

page 12, line 15-17: What is the measure that indicates deep convection on 3rd of December 2017?

page 12, line 25-29: What is the line of argumentation here? The text deals initially with COSMO-DE and its graupel scheme. Afterwards, we learned something about reduced precipitation over the coastal Arabian and then, downscaling issues of the UM are referred. There is no logic at all.

page 12, line 33-34: What do you mean with that sentence? What means necessary?

page 13, line 4-5: The english text reads strange starting from 'where little ...'.

page 13, line 6-8: The author is telling about tuned parameterizations in NWP models, in particular for specific resolutions and scales. The study presented here should give valuable insight into the treatment of convection and the impact on precipitation. I am not convinced at all that we learn with this study something new and not known from former studies. We learn about model results from the storm 'OCKHI' nothing more. This study does not help to conclude about what are the problems with dynamical downscaling nor at which resolution to switch off parts of the parameterization of convection.

**4.6 Figures and Tables**

figure 1: Do we need it? The text explains everything one needs.

figure 2: Which simulation is shown regarding the CAPE? The extend of the COSMO domain is not large enough to extract that information.

figure 3: Which experiment of COSMO is shown? (CPC)

figure 4: The CAPE observation at 00UTC of 3.12.2017 and at 69.15 degree East and 11.82 degree North should be marked in each plot.

figure 5: Which area is taken for averaging?

figure 6: Which area is taken for averaging? What is meant with the last sentence in the caption?

table A1: The version number of COSMO is missing. Reference for grid-scale precipitation, vertical turbulence diffusion and surface-layer turbulent fluxes is missing.

table A2: Which time is analyzed? The position at 00UTC of 3.12.2017? Why is the analysis not done for other stages of the storm?

---

## Author Comment (AC1) · 10 Sep 2019

article [a4paper,top=1.0in,bottom=1.0in,left=1.0in,right=1.0in]geometry fancyhdr setspace color hyperref graphicx textcomp soul Response to the Reviewer's Comments: NHESS-2018-288

**Impact analysis of dynamical downscaling on the treatment of convection in a regional NWP model - COSMO: a case**

**study during the passage of a very severe cyclonic storm "OCKHI"**

By:

**Roshny S., D. Bala Subrahamanyam, Anurose T. J. and Radhika Ramachandran**

Ms Reference No.:NHESS-2018-288

====================================

**Summary of Revision (to the Editorial Board, NHESS)**

Dear Dr. Fabrizio Masci,

On behalf of myself and my co-authors, I would like to extend my sincere thanks to you and your supporting Editorial team for your efforts in evaluation of our manuscript. We would also like to place on records our sincere appreciation to Dr. Ronny Petrik and other anonymous reviewer for their valuable comments and suggestions, which have helped us in extending the scope of paper and improving the quality of scientific content of our manuscript. We have addressed almost all the suggestions/queries raised by both the reviewers and have made necessary modifications in the manuscript.

After incorporating reviewer's suggestions, the revised manuscript includes fine-resolution (i.e., 0.25° grid spacing) ERA5 and NCEP FNL reanalysis fields for assessment of initial conditions and validation of CAPE and other meteorological fields simulated through different numerical experiments of COSMO. We have also

included satellite-based IMERG precipitation measurements (available at 0.10°
horizontal resolution) for validation of rainfall simulations. As per the suggestions of
both the reviewers, numerical experiments with COSMO model are also re-designed.
For investigation of the vertical structure of cyclonic storm, a new figure of vertical
cross section of equivalent potential temperature is also included in the Results and
Discussion.

Having addressed most of the queries/suggestions pointed out by the reviewers, we
are now quite optimistic that you will find the revised version of our manuscript ac-
ceptable for publication in NHESS Journal. Point-to-Point Response to the Reviewer's
comments and summary of modifications carried out in the revised manuscript is
attached as an Appendix to this letter.

Thanking You,

Dr. D. Bala Subrahamanyam
Corresponding Author
(On behalf of all the co-authors)

Dated: September 05, 2019.

**POINT-TO-POINT RESPONSE & SUMMARY OF REVISION**

REVIEWER#1

Comments from Referee

*The study evaluates the representation of a cyclone over the Arabian Sea in COSMO model simulations at different horizontal resolutions and different treatments of convection. More specifically, the authors performed simulations at a grid spacing of (i) 0.0625° with a parameterized convection, (ii) 0.025° also with a convection scheme, and (iii) 0.025° with explicit convection. Precipitation and CAPE fields from COSMO are then compared to ERA-Interim reanalysis data in an attempt to evaluate which model configuration better represents the convection and precipitation during the passage of the cyclone over the Arabian Sea along the Indian Peninsula.*

*Overall, I found the study potentially relevant and the manuscript carefully written. However, there are several major flaws in the model setup of the experiments, the meteorological evaluation, and the choice of the data set that serves for the model evaluation. Therefore, a substantial part of the analysis is invalid and the conclusions remain unsubstantiated. In my view, the required corrections go beyond major revisions. However, the study has much potential when these major comments are accommodated. Tropical cyclones cause frequently severe socioeconomic impacts, and their simulation and predictability are of high interest to the scientific community and operational weather forecasting. Also, the model experiments with different representations of convection are relevant and deserve attention. Therefore, I would like to encourage the authors to implement these changes and to resubmit the manuscript.*

Author's Response

We would like to thank the reviewer for endorsing the potential relevance of our manuscript ("I found the study ...carefully written."), and also agree with the objections raised thereafter. During the revision of our manuscript, we have modified the numerical experiments with COSMO by replacing the DPC simulations with a new set of simulations, wherein the grid resolution of COSMO is kept at 0.0625° and convection parameterization scheme is switched off. Furthermore, for meteorological evaluation of our model simulations, we have now included fine-resolution ERA5 and NCEP FNL Reanalysis fields (available at a grid resolution of 0.25°). For validation of rainfall simulations, we have made use of satellite-based IMERG (Integrated Multi-satellitE Retrievals for Global precipitation measurement) observations. Furthermore, the model domain of COSMO is also extended to a larger area over the Arabian Sea which covers the entire track of OCKHI storm, and all the simulations are carried out for new domain. After incorporation of new datasets, with re-designed numerical experiments, the Results and Discussions as well as our Conclusions are substantially improved.

Author's changes in the manuscript

- **Introduction:** Scope of the manuscript is revised.

- **Data:** Details of ERA5, NCEP FNL Reanalysis and IMERG observations are added.

- **Numerical Experiments in the COSMO Model:** DPC Simulations are eliminated and are replaced with CNC simulations (0.0625° grid resolution, and convection scheme switched off).

- **Figures**: Figure 5 and Figure 7 are eliminated. One new figure is included to show the sea level pressure and wind vectors from reanalysis fields and con-

current simulations from COSMO. Furthermore, one more figure is added for representation of vertical cross section of equivalent potential temperature along the latitudes.

- As an outcome of the above-mentioned modifications in the manuscript, **Results and Discussion**, and **Conclusions** sections are also substantially revised.

Comments from Referee

**1. Dynamical Downscaling:**

*At several occasions, including the title, the abstract and the conclusions, the manuscript claims to assess the impact of dynamical downscaling on the representation of convection in the numerical weather prediction model COSMO. Dynamical downscaling refers to the use of a limited area model or regional climate model to provide more detailed information on weather or climate that cannot be provided by a global weather or climate model that typically has a relatively coarse model resolution. When claiming that the impact of dynamical downscaling is assessed, the study would need to show a comparison between the global data (from ICON) and the limited area model (COSMO), for example, for the simulation of precipitation and CAPE. The present manuscript does not show any data from the global model. In order to evaluate impacts of dynamical downscaling, as the manuscript claims to address, the authors would need to include a comparison between the ICON and COSMO data, and then to use observation-based data to show that the dynamical downscaling indeed leads to an improvement.*

*At other places in the manuscript, for example in Section 4, the text seems to imply that the use of different horizontal grid spacing aids the evaluation of the impacts of dynamical downscaling. This is incorrect as it can only help to investigate the sensitivity of convection to the model resolution. In other words, the use of different horizontal model resolutions is not the same as dynamics downscaling.*
[Figure]

Author's Response

We agree with the reviewer's suggestions about the "dynamical downscaling". In the revised manuscript, we have included meteorological fields of CAPE, sea level pressure and wind vectors from ICON global model. These fields are later compared with the COSMO simulations at dynamically downscaled finer grids. Furthermore, we have included satellite-based precipitation measurements for validation of rainfall simulations.

We also agree with the reviewer about rephrasing of sentences in Section 4 dealing with the dynamical downscaling. In this section, we have explicitly mentioned that the present work deals with the sensitivity of model's grid resolution to the convection parameterization scheme.

Author's changes in the manuscript

- **Data:** Details about ICON global model and other reanalysis fields are included.

- **Results and Discussions:** CAPE, sea level pressure and wind vectors extracted from global reanalysis fields are compared with COSMO model simulations. Similarly, rainfall simulations of COSMO are validated against the satellite-based IMERG observations.

- As an outcome of the above-mentioned modifications in the manuscript, **Results and Discussion**, and **Conclusions** sections are also substantially revised.

Comments from Referee

*2. Observations for model validation and comparison*
*The study uses the ERA-Interim data set from the ECMWF as a means to validate*

*the COSMO model simulations. This approach is problematic since the ERA-Interim reanalysis is produced at a resolution of about 0.7° x 0.7°, much coarser than the COSMO model simulations which use a horizontal grid space of about 3-7 km. Later, in the analysis it becomes indeed clear that the data is much coarser (e.g., page 8, lines 26-27) and that the center of the cyclone is more off in the ERA-Interim data than in the COSMO simulations. As a consequence, the comparison between the COSMO simulation experiments and ERA-Interim as shown in Figures 5 and 7 is not relevant. Therefore, the conclusions based on this comparison, as for example phrased in the last sentence of the abstract, are not supported by a valid analysis.*

*I highly recommend to use precipitation observations based on satellite estimates, for example TRMM (Huffmann et al., 2007) or any other satellite product, whereas CAPE fields from the operational IFS analysis from ECMWF may provide higher resolution data than the ERA-Interim reanalysis.*

Author's Response

AGREED AND IMPLEMENTED. CAPE measurements from fine-resolution global re-analysis fields (ERA5 and NCEP FNL, both with 0.25° grid resolution) are used in the revised figures. We also accept to make use of precipitation observation based on satellite estimates. In this regard, we would like to mention that TRMM observations over the oceanic regions for the period of OCKHI storm are not available. Hence, as an alternative option, we have used satellite-based IMERG precipitation measurements, which are available at 0.10° grid resolution and are widely used for the precipitation studies.

Author's changes in the manuscript

- **Figures:** CAPE fields from ERA5 and NCEP FNL reanalysis are shown in the revised figures.

- Furthermore, IMERG satellite-based precipitation measurements are used for depicting the observed 24 h accumulated rainfall.

- Accordingly, the write-up describing the modified figures is also revised.

Comments from Referee

**3. Model domain**
*The used model domain is very small with only 10 degrees / 1000 km in zonal and meridional direction. In fact, the cyclone is located near the boundary of the domain at the initial time of the simulations with a +48 and + 36 hour lead time, as shown in Figure 2. This model configuration is problematic for obtaining proper results. I recommend using a model domain that is sufficiently covers the tropical cyclone throughout the simulation. As also written in section 5.1.2 (page 11 and lines 30-32), it is understood that computational resources can be a limitation; however, this cannot justify a model simulation that does not support a valid study.*

Author's Response

AGREED AND IMPLEMENTED.

[Figure]

Author's changes in the manuscript

- COSMO domain is enlarged over the Arabian Sea (6.0° N to 22.0° N; and 66.0° E to 82.0° E).

Comments from Referee

**4. Simulation experiments**
*The study uses three different simulation experiments; (1) with a grid spacing of 0.0625°(~ 7 km) and convection parameterized, (2) with a grid spacing of 0.025°(~ 3 km) and convection parameterized, and (3) with a grid spacing of 0.025°(~ 3 km) and without convection scheme. Following previous studies, convection schemes can potentially be switched of at the order of a 7 km grid spacing, whereas convection may largely be resolved when using a grid spacing of 3 km (e.g., Marsham et al., 2013). This is also explicitly stated in the manuscript on page 7, lines 17-18. The results show that experiment (2) does not add much to experiment (1), whereas the convection-permitting simulation of experiment (3) shows a lot of details as compared to experiment (2). Therefore, I would recommend to replace experiment (2) by a simulation with a grid space of 0.0625°(~ 7 km) and without the use of a convection scheme.*

Author's Response

AGREED AND IMPLEMENTED. DPC Simulations are replaced with CNC (Control simulations, with No Convection parameterization) simulations, wherein the grid resolution of COSMO is kept as 0.0625°, and the convection parameterization scheme is switched off.

Author's changes in the manuscript

- **Numerical Experiments in the COSMO Model:** Necessary changes are made.

Comments from Referee

**5. Meteorological analysis**
*The analysis in the manuscript is limited to CAPE and precipitation. In order to learn more about the representation of the tropical cyclone in the different model simulations, it may be helpful to extend the analysis with other meteorological variables, for example, sea level pressure and equivalent potential temperature. In particular, SLP can reveal information about the track of the cyclone, which can be compared to the observations and thus be a measure for the accuracy of the different model simulations.*

Author's Response

AGREED AND IMPLEMENTED. Analysis of different meteorological fields is now extended to sea level pressure, wind vectors, and equivalent potential temperature. Contour maps of sea level pressure and wind vectors are used for the determination of cyclone track, whereas a vertical cross-section of equivalent potential temperature map across latitudes is included as a new figure for understanding the vertical structure of the cyclonic storm.

Author's changes in the manuscript

- **Figures:** New figures of sea level pressure, wind vectors, and vertical cross section of equivalent potential temperature are added.
Comments from Referee

*In addition, the area averaged amounts of precipitation and CAPE, as shown in Figures 5 and 7, may not be appropriate for a comparison of the model simulations to observational data. The area averaging leads to a loss of detailed information and can be misleading as a comparison.*

Author's Response

AGREED. These figures are eliminated.

Author's changes in the manuscript

• Old Figure 5 and 7 are eliminated.

Comments from Referee (Minor Comments:)

Author's Response and Changes in the manuscript

Below we present a summary on all the minor comments raised by the reviewer (*Italic Letters*), and our response/changes in manuscript just beneath the reviewer's comments. Overall, we have taken care of all the minor comments in the revised version of manuscript.

• *At several places (e.g., lines 2, 5 and 8 on page 3), the text speaks about a cyclone. Is there a specific reason why not to speak about a tropical cyclone? The term "cyclone" is a very general term that also covers extratropical cyclones that*

*are found in the extratropics.*

AGREED AND REPLACED. Wherever relevant, we have replaced the word "cyclone" with "tropical cyclone" in respective sections.

- *Page 1, lines 19-20. Instead of speaking about "the smallest and most compact weather processes", please, speak in terms of spatial and time scales.*

AGREED AND CORRECTED. Necessary modifications are done in this sentence by incorporating the spatial and time-scales of convective processes.

- *Please, omit lines 17-18 on page 2; "With the availability ... becoming finer.". The use of higher model resolutions is primarily limited by computational resources, not by the availability of observations.*

AGREED AND OMITTED. The above sentence is omitted.

- *At several places in the manuscript (e.g., page 3, lines 10-11, page 8, line 9, and page 12, lines 3-4) the text speaks about simulation experiments with different initial conditions. This can be understood by readers as use of slightly perturbed initial conditions at the same date, as for example used for ensemble simulations. Instead, the difference between the simulations are different forecast lead times with respect to the episode of interest. Please, rephrase the text where needed.*

AGREED AND IMPLEMENTED. Above sentences are rephrased and corrected.

- *Section 2. Please, specify which COSMO version is used.*

AGREED AND INCLUDED. Version 5.05 of COSMO is used for simulations in the present study.

- *Page 4, lines 12-13 The phrase "Since the convective processes ... ... much smaller than those resolved by mesoscale and regional NWP models" is not entirely correct. In case of high-resolution simulations, for example, with a horizontal*

*grid spacing of 3 km, convective processes may be largely resolved by the model.*

AGREED AND CORRECTED. Above sentences are rephrased and corrected.

- *Page 5, lines 8 "to the standard version of the model"; what is the standard version of the model in their study? Please,clarify in the text.*

AGREED AND INCLUDED. The standard version of the Community Atmosphere model Version 2 (CAM2) employs the deep convective scheme of Zhang and McFarlane (1995). The authors of the cited paper have used Tiedtke scheme (1989) revised by Nordeng (1994) for their study. These details are included in the revised manuscript.

- *Section 3.1 Please, state in the text how many model levels are used for the model simulations.*

AGREED AND INCLUDED. We have configured the COSMO model simulations with a total of 50 vertical levels. This information is included in the revised manuscript.

- *Page 5, line 28 speaks about the use of ICON global model data for a period of 9 days. This is not consistent with the ICON model data that are used for four different time instances on 2 days as mentioned later in the text (Page 6, line 10). Please, correct.*

AGREED AND CORRECTED. Above sentences are rephrased and corrected.

- *Page 6, lines 13-18. Please, omit the technical description "COSMO model ... ... fine-grid resolution."*
AGREED AND OMITTED. Above sentences are omitted, and we have included citation of the model's users guide for further technical details.

- *Page 7, lines 10-12. Why don't you show the precipitation from ICON? See also major comment number 1.*

AGREED AND IMPLEMENTED. As mentioned above, now the revised manuscript include satellite-based IMERG precipitation measurements for validation. Hence, we are directly showing the precipitation measurements from IMERG rather than reanalysis or analysis fields.

- *Page 7, line 26. In what way was the tropical cyclone rare? In its intensity, duration and / or socioeconomic impacts? Please,be specific.*

  AGREED AND IMPLEMENTED. The OCKHI cyclonic storm was categorized as a Very Severe Cyclonic Storm (VSCS), which was formed in the month of December. Historically speaking, none of the Depressions or Cyclonic Storms formed over the Comorin Sea in the month of December ever became a VSCS in the last 100 years or so. Secondly, this storm attained the status of a Cyclonic Storm from the stage of Depression within 6 h. Its rapid intensification was yet another rare event which was extremely unusual. These details are included in the revised manuscript.

- *Page 8, lines 12-13. Please, state the source of these precipitation observations.*

  AGREED AND INCLUDED. These observations were cited from the IMD report on OCKHI. Appropriate citations are included in the revised manuscript.

- *I recommend to restructure sections 5.1.1 as 5.2 and 5.1.2 as 5.3.*
  AGREED AND RE-NUMBERED. Section numbering is corrected accordingly.

- *Page 9, line 4. It is not only the state of the lower troposphere that defines CAPE. Please, replace "lower atmosphere" by "the thermodynamic conditions".*
  AGREED AND REPLACED.

- *Page 9, lines 6-7. Please, clarify which forecast step is used from ERA-Interim.*

  AGREED AND INCLUDED. CAPE fields are corresponding to 00 UTC of 03 December 2017.

- *Page 9, lines 13-14 "The ECMWF fields were almost off by more than 100 kms from ..." and page 10, lines 12-13 " ... the CAPE magnitudes obtained from ECWMF fields were always overestimated ..." and page 10, lines 24-29 "In this regard ... ... a smaller mesoscale region only". This shows that the ERA-Interim data is not suitable for validation of the model simulations, see also major comment number 2.*

  AGREED. We have included fine-resolution global data of ERA5 and NCEP FNL reanalysis with ERA-Interim data.

- *Page 10, line 32. Are these precipitation amounts per day?*

  AGREED AND INCLUDED. Yes, these precipitation amounts are for 24 h between 00 UTC of 02 December 2017 to 00 UTC of 03 December 2017. These details are included in the revised manuscript.

- *Page 11, lines 2-3. I cannot follow the sentence "Even if ... accumulated rainfall". Please, rephrase.*

  AGREED AND REPHRASED. This sentence has been rephrased in the revised manuscript.

- *The conclusions at page 11, lines 19-20 "However, switching off ... ... rainfall over the Arabian Sea." and at page 12, lines 12-13 "Fine representation ... ... accumulated rainfall magnitudes." are invalid due to the comparison of COSMO simulations to ERA-Interim data. Satellite-based estimates may provide a base for a more realistic and useful comparison, see also major comment number 2. Moreover, Figure 6 shows that the DNC simulation has intense rainfall, although area-averaged amounts as in Figure 7 may be lower as compared to ERA-Interim.*

  AGREED AND CORRECTED. We have included satellite-based IMERG precipitation measurements for validation of rainfall simulations. Thus, these points are

well-addressed in the revised manuscript.

- *Page 12, lines 30-31. The sentence "There is a visible increase ... ... over the tropical oceans" needs to be supported by references or otherwise be removed.*

  AGREED AND REMOVED. The above sentence is removed.

Comments from Referee (Writing Comments:)

Below we present a summary on all the "Writing Comments" raised by the reviewer (*Italic Letters*), and our response/changes in manuscript just beneath the reviewer's comments. Overall, we have taken care of all these comments in the revised version of manuscript.

- *Page 1, line 5. Please, consider to replace "inter-linking of" by "interplay between".*

  AGREED AND REPLACED.

- *Page 1, line 15. Please, replace "an NWP" by "a NWP" and write out NWP. Abbreviations used in the abstract need again to be defined within the text of the manuscript upon first use.*

  AGREED AND CORRECTED.

- *Page 1, line 20 – Page 2, line 1. Please, replace "surface to the troposphere" by "surface to the upper troposphere"*

  AGREED AND REPLACED.

- *Page 2, line 3. Please, replace "Convection process" by "Convective processes" or "Convection".*

  AGREED AND REPLACED.

- *Page 2, lines 9-10. Please, rephrase the sentence "However, the process of con-vection ... interaction with radiation", for example, as "Moreover, convection in-volves complex interactions with cloud formation which influence the atmospheric circulation through radiative effects." or in a similar direction.*

  AGREED AND REPHRASED.

- *Page 2, line 12. Please, replace "inter-linked" by "complex".*

  AGREED AND REPLACED.

- *Page 2, line 13. Please, replace "Further" by "Furthermore, ".*

  AGREED AND REPLACED.

- *Page 2, line 16. Please, replace "is apparently inter-linked with" by "constrained by" or in a similar direction.*

  AGREED AND REPLACED.

- *Page 2, line 19. Please, replace "As on today" by "At present" or "Currently".*

  AGREED AND REPLACED.

- *Page 2, line 26. Please, remove "physical and".*

  AGREED AND REMOVED.

- *Page 3, lines 16-17. Please, correct the sentence by writing "COSMO (formerly known as ... ...in Switzerland) is a non-hydrostatic ... ... model that was initially ...".*

  AGREED AND CORRECTED.

- *Page 4, line 9. Please, replace "the conserved framework" by "the conservation of" and in line 10, replace "Different schemes" by "Schemes".*

AGREED AND REPLACED.

- *Page 4, line 14. Please, replace "resolved" by "estimated" or "represented".*
AGREED AND REPLACED.

- *Page 4, line 18. Please, replace "cumulus" by "convective".*
AGREED AND REPLACED.

- *Page 4, line 21. Please, replace "the Convectively" by "Convective".*
AGREED AND REPLACED.

- *Page 4, line 25. Please, replace "COSMO model" by "The COSMO model".*
AGREED AND REPLACED.

- *Page 4, line 30. Please, replace "lies" by "lie".*
AGREED AND REPLACED.

- *Page 5, line 12. Please, remove the comma after "that".*
NO CHANGES ARE MADE. We could not find the above mistake in original manuscript.

- *Page 5, line 11. Please, replace "can provide improved" by "can improve".*
AGREED AND REPLACED.

- *Page 5, line 31. Please, replace "form" by "from".*
NO CHANGES ARE MADE. The above sentence is correct to the best of our knowledge, and "form" is correctly used, hence no changes are made.

- *Page 6, line 11. Please, remove "and meteorological".*
AGREED AND REMOVED.

- *Page 6, line 27 as well as on page 7, line 8. Please, replace "under the framework of" by "using the" or "with the".*

  AGREED AND REPLACED.

- *Page 7, line 1, 10, and elsewhere in the manuscripts. Always, use a comma, before "respectively".*

  AGREED AND IMPLEMENTED THROUGHOUT THE MANUSCRIPT.

- *Page 7, lines 2-3. Please, remove "to the COSMO model" and write "of the actual episode".*

  AGREED AND IMPLEMENTED.

- *Page 7, lines 6. Please, remove "to the COSMO model".*

  AGREED AND REMOVED.

- *Page 7, lines 15-17. Please, rewrite this long sentence. Stating that you switched off the convection scheme or use a convection-permitting simulation is sufficient.*

  AGREED AND RE-WRITTEN.

- *Page 7, line 14. Please, rephrase "are treated directly" by "are explicitly simulated", or "permitted" or in that direction.*

  AGREED AND REPHRASED.

- *Page 7, lines 26 and 31. Please, rephrase "residence time" by "life span" of "life cycle".*

  AGREED AND REPLACED.

- *Page 7, line 30. Replace "between" by "from".*

  AGREED AND REPLACED.

- *Page 7, line 30 and at many other places in the manuscript. Rewrite "01st December" and "5 th December" as "1 December" and "5 December".*

  AGREED AND IMPLEMENTED THROUGHOUT THE MANUSCRIPT.

- *Page 8, line 3. Please, replace "The CAPE" by "CAPE".*

  AGREED AND REPLACED.

- *Page 8, line 7. Please, reverse "final dissipation and landfall".*

  AGREED AND CORRECTED. Sentence is rephrased and is rewritten as "landfall and final dissipation".

- *Page 8, line 22. Please, replace "corresponding to" by "on".*

  AGREED AND REPLACED.

- *Page 9, line 5. Please, replace "proactive" by "conducive" or "favorable".*

  AGREED AND REPLACED.

- *Page 9, line 10. Please, remove ", and the category of storm was retained as VSCS.".*

  AGREED AND REMOVED.

- *Page 9, line 18. Please, replace "downscaling of" by "a higher".*

  AGREED AND REPLACED.

- *Page 9, line 20. Please, replace "which was actually not true for" by "occurred on".*

  AGREED AND REPLACED.

- *Page 9, lines 24. Please, write "Figures ... depict...".*

AGREED AND CORRECTED.

- *Page 9, lines 27-28. Please, replace "was" by "were" and "activities" by "activity".*
  AGREED AND REPLACED.

- *Page 10, line 15. Define the abbreviation "CS" or write full out.*
  AGREED AND CORRECTED. All abbreviations are once again carefully checked.

- *Page 10, line 18. Remove "got" and "it got".*
  AGREED AND REMOVED.

- *Page 10, line 19. Replace "reasonably" by "substantially".*
  AGREED AND REPLACED.

- *Page 11, lines 9-11. Please, replace "requirements" by "experiments", "amount" by "amounts", "is" by "are", "least" by "shorter".*
  AGREED AND REPLACED.

- *Page 12, lines 4-5. Please, replace "in determination of" by "to determine".*
  AGREED AND REPLACED.

Comments from Referee (Figures and Tables)

- *Figure 1. Please, remove the words downscaling in the caption as all simulations are downscaled in the sense that the simulations are fed by global data. Instead, indicate the resolution of horizontal grid spacing of 7 and 3 km.*
  NECESSARY CORRECTIONS ARE DONE IN THE FIGURE CAPTION.

- *Figure 2. The caption speaks about CAPE from COSMO. CAPE fields extend outside the model domain that is indicated by the black box when I understand correctly. Is this perhaps CAPE from ERA-Interim?*

  AS THE MODEL DOMAIN IS EXPANDED, THIS FIGURE ITSELF IS REVISED.

- *Figure 6. Is this the accumulated rainfall in the 24 hours prior or after 00 UTC, 3 December 2017? Please, clarify.*

  FIGURE CAPTION IS CORRECTED AND ABOVE DETAILS ARE INCLUDED.

- *Table A2. For the sake of consistency, I would recommend to also include the results from the CPC simulation.*

  CPC SIMULATION DETAILS ARE ALSO INCLUDED IN THE TABLE.

---

## Author Comment (AC2) · 10 Sep 2019

article    [a4paper,top=1.0in,bottom=1.0in,left=1.0in,right=1.0in]geometry    fancyhdr setspace color hyperref graphicx textcomp soul    Response to the Reviewer's Comments: NHESS-2018-288

[Figure]

**study during the passage of a very severe cyclonic storm "OCKHI"**

By:

Roshny S., D. Bala Subrahamanyam, Anurose T. J.
and Radhika Ramachandran

Ms Reference No.:NHESS-2018-288
=====================================

**Summary of Revision (to the Editorial Board, NHESS)**

Dear Dr. Fabrizio Masci,

On behalf of myself and my co-authors, I would like to extend my sincere thanks to you and your supporting Editorial team for your efforts in evaluation of our manuscript. We would also like to place on records our sincere appreciation to Dr. Ronny Petrik and other anonymous reviewer for their valuable comments and suggestions, which have helped us in extending the scope of paper and improving the quality of scientific content of our manuscript. We have addressed almost all the suggestions/queries raised by both the reviewers and have made necessary modifications in the manuscript.

After incorporating reviewer's suggestions, the revised manuscript includes fine-resolution (i.e., 0.25° grid spacing) ERA5 and NCEP FNL reanalysis fields for assessment of initial conditions and validation of CAPE and other meteorological fields simulated through different numerical experiments of COSMO. We have also

included satellite-based IMERG precipitation measurements (available at 0.10° horizontal resolution) for validation of rainfall simulations. As per the suggestions of both the reviewers, numerical experiments with COSMO model are also re-designed. For investigation of the vertical structure of cyclonic storm, a new figure of vertical cross section of equivalent potential temperature is also included in the Results and Discussion.

Having addressed most of the queries/suggestions pointed out by the reviewers, we are now quite optimistic that you will find the revised version of our manuscript acceptable for publication in NHESS Journal. Point-to-Point Response to the Reviewer's comments and summary of modifications carried out in the revised manuscript is attached as an Appendix to this letter.

Thanking You,

Dr. D. Bala Subrahamanyam
Corresponding Author
(On behalf of all the co-authors)

Dated: September 05, 2019.

**POINT-TO-POINT RESPONSE & SUMMARY OF REVISION**

REVIEWER#2

Comments from Referee

**1. General Comments**
*The paper reviewed is about a very severe cyclonic storm in the Arabian Sea. In the framework of a downscaling experiment, the author investigates the impact of model resolution and convective parameterization on the results. The English language in the text is IMO proper to achieve a good flow of reading and to get the context right. The structure is clear and the figures and tables are well done. However, main issues appear with the text which call at least for a major review. If the author is not adequately tackling that issues, the scientific content will still be questionable (i.e. very likely a rejection of the content).*

Author's Response

We thank Dr. Ronny Petrik for his critical appraisal of the manuscript and numerous suggestions. We have taken into account of all the suggestions, and have revised our manuscript substantially by addressing all queries raised by both the reviewers.

Author's changes in the manuscript

- Below, we provide Point-to-Point Response to all the queries and concerns raised by REVIEWER#2.
Comments from Referee

**1.1 The authors intention - analysis of sensitivity for initial conditions**

*In the paper presented a sensitivity to lead times is done. However, to identify sensitivity for initial conditions, other forcing data have to be considered. In the case of the Arabian sea I would prefer ERA-analysis, ERA5-reanalysis, MERRA2 reanalysis or NCEP analysis as well as reanalysis. Thus, having three different types of analysis, the sensitivity study is much more convincing. One would incorporate the spread originating from the different physical parameterization schemes and the assimilation techniques.*

Author's Response

We agree with the reviewer's suggestions, and accordingly we have included ERA5 and NCEP FNL reanalysis fields in the revised manuscript. Different meteorological parameters, such as: CAPE, sea level pressure, and wind vectors extracted from these reanalysis data, are compared with the COSMO model simulations. Also an inter-comparison of these global fields is carried out for the assessment of initial conditions. Since, our primary goal of this research article is confined towards assessment of convection parameterization scheme at different horizontal grid resolution, we have not extended the scope of this article towards data assimilation, or the spread originating from different physical parameterization schemes. Nevertheless, we have adequately rephrased the write-up in the revised manuscript to make the scientific contents of this article well-focused.

Author's changes in the manuscript

- **Data:** Details of ERA5, NCEP FNL Reanalysis and IMERG observations are added.

- **Results and Discussion:** Details on meteorological fields, such as: CAPE, sea level pressure, and wind vectors from reanalysis fields are discussed in line with the COSMO model simulations.

- **Figures:** Above-mentioned meteorological fields are included in the revised figures.

Comments from Referee

*1.2 The authors intention - analysis of sensitivity for parameterization of convection*

*From previous studies it is already clear that parameterization for deep convection can be switched-off for resolutions smaller than about 3-5 km. The interesting question is the 'about'. Therefore, I see no reason why to add the DPC experiment with 2.8 km resolution. However, the sensitivity analysis would get more meaningful if the author decides:*

- *deal with an experiment in the convective 'grey zone' and performs a simulation at 4 to 5 km with parameterization for convection switched-off and switched-on.*

- *to investigate the need for parameterization of shallow convection. That means to add a simulation at 2.8 km resolution deactivating the shallow convection (which is active in the standard configuration).*

*To clarify, the recent content of the paper is somehow '2.8 km resolution leads to more details in the CAPE and precipitation fields, compared to CPC experiment with 7 km resolution. The experiment DPC is unnecessary because the patterns are smoothed and the area-averaged precipitation is the same as for CPC. The CAPE values are off compared to ERA-Reanalysis and DNC, CPC.' However, addressing the research questions the author mentioned in the introduction, it is required to go beyond the experiments introduced in the recent version of the paper.*

Author's Response

We agree with the reviewer's suggestions. DPC Simulations are replaced with CNC (Control simulations, with No Convection parameterization) simulations, wherein the grid resolution of COSMO is kept as 0.0625°, and the convection parameterization scheme is switched off. Furthermore, we have made use of fine-resolution reanalysis data from ERA5 and NCEP FNL for comparison of CAPE, and other meteorological fields. In addition to this, we have also utilized satellite-based precipitation measurements in the revised manuscript for validation of rainfall simulations.

We also appreciate the reviewer's suggestion on "the need for parameterization of shallow convection", however the current version of COSMO offers simulation of convection as per Tiedtke parameterization scheme. Hence, we have confined our analysis to the sensitivity of Tiedtke convection parameterization scheme to horizontal grid resolution during the passage of OCKHI storm. We will definitely try to address the second aspect in a more detailed manner in a separate research article, as inclusion of all these stuff may be beyond the scope of present research article.

Author's changes in the manuscript

- **Numerical Experiments in the COSMO Model:** We have eliminated the DPC simulations, which are now replaced with a new set of simulations, namely - CNC.

- **Figures:** New figures are drawn with new reanalysis data, as well as new simulation experiment.

Comments from Referee

**2. Evaluational Basis**
*The evaluation of the results is superficial. First, ERA-reanalysis data are not helpful in measuring the quality of the high-resolution model. The author should consider satellite data from TRMM as remote sensing observations. In addition, the data from IMD are referred but at no time a quantitative comparison is provided to the reader. Without such a comparison, the author cannot raise arguments like 'the downscaling did not improve rainfall prediction' or 'the CAPE magnitudes obtained from ECMWF fields were always overestimated'. The basis for evaluation could be more improved by incorporating radiosonde data or satellite data about the cloud structures. The ERA-reanalysis can be useful for qualitatively analyzing those meteorological parameters, which are more or less instantaneously assimilated, as the mean sea level pressure.*

Author's Response

We agree with the reviewer's views about limitations in the ERA-Interim reanalysis fields. For making the comparison of COSMO simulations with reanalysis fields more meaningful, we have now included fine-resolution ERA5 and NCEP FNL reanalysis data, which are available at 0.25° grid resolution. Furthermore, we have also included

satellite-based precipitation measurements from IMERG for validation of 24 h accumulated rainfall.

Author's changes in the manuscript

- **Results and Discussions** and **Figures:** We have redrawn all the figures with inclusion of new datasets (ERA5 and NCEP FNL Reanalysis, and IMERG precipitation measurements). Accordingly, associated write-up is also modified.

Comments from Referee

**3. Robustness of the Analysis**
*The author confines himself to the analysis of precipitation and CAPE. Much more meteorological parameters have to be evaluated to get a clue about the differences in the model results and the related performances. It would be very beneficial to study the vertical structure of the cyclone along the path or as a cross section, to visualize the path of the eye (distance to observed position) for all configs in one figure, to look at the cloud structures, the simulated vertical velocity and the vertical integrated cloud content as well as moisture-flux divergence (as a precursor for the convection parameterizations). Furthermore, the idea of downscaling is to add some value to the forcing model, which is the ICON in your case. The author misses to analyze which of the configurations is superior over the forcing simulation. It is not fair and not useful to compare the high resolution simulations with a global reanalysis, which cannot hold as a reference for a 'global prediction' as well as an observational field. It is much too coarse compared to the models the author deals with.*

*In addition, I am asking myself why not to choose a model domain capable of resolving the initiation of the storm. i.e. the extension of the domain in Southern direction by 1 degree and in Eastern direction by 2 degrees captures the whole intensification stage*

*of the storm. Doing so, one gets more independent from the global forcing regarding lateral conditions.*

*However, it is still a big challenge to extract some general scientific implications from a single case study for the scientific community. Thus, it would be worth to look at other comparable events which would extend the study in a reasonable manner and which would result in a more robust statistical and scientific basis.*

Author's Response

We have extended the scope of our revised research articles beyond the precipitation and CAPE. Now, the mean sea level pressure, wind vectors and vertical cross-section of equivalent potential temperature along the latitudes is also included in the Results and Discussion.

We have included fine-resolution global reanalysis data (ERA5 and NCEP FNL Reanalysis, available at 0.25° grid resolution), and satellite-based IMERG precipitation measurements (available at 0.10° grid resolution) for validation of COSMO simulations.

COSMO model domain is extended to a larger area for covering the entire track of OCKHI storm.

Finally, we agree with the reviewer that inclusion of more comparable events would extend the robustness of results in a statistical and scientific basis. However, as we mentioned in the title itself, the present work is undertaken as a case study wherein the sensitivity of convection parameterization scheme on the grid resolution, and its impact on forecast fields with designing different numerical experiments, is investigated. Nevertheless, we take the reviewer's comments in a positive manner and will carry similar studies in future to include more number of cyclonic storms.

Author's changes in the manuscript

- **Results and Discussion:** Sea level pressure, wind vectors, and equivalent potential temperature are newly included meteorological fields.

- **Figures:** A new figure showing the vertical cross-section of equivalent potential temperature across latitudes is added for the investigation of vertical structure of the cyclonic storm.

Comments from Referee (4. Specific Comments)

Below we present a summary on all the Specific Comments raised by the reviewer (*Italic Letters*), and our response/changes in manuscript just beneath the reviewer's comments. Overall, we have taken care of all these comments in the revised version of manuscript.

- ***4.1 Introduction***

  *The introduction is well written and with a nice literature review. However, it is too general, i.e. a literature discussion about tropical storms is missing as well as the performance of models resolving them. Furthermore, I miss a section overview at the end.*

  AGREED AND INCLUDED. We have included details on performance of models resolving storms, as well as an overview of the manuscript.

- *page 1, line 17: 'to name a few' can be skipped*

  AGREED AND SKIPPED.

- *page 2, line 1: Start a new sentence*

  AGREED AND CORRECTED.

- *page 2, line 21: 'meteorological data' can be replace by data. The forcing data are much more than meteorological data (hydrological, ...)*

  AGREED AND REPLACED.

- *page 2, line 31: Regarding the discussion of resolution needed to achieve a complete explicit representation of convection, the author should refer Bryan (2003) [Resolution Requirements for the Simulation of Deep Moist Convection].*

  AGREED. Bryan (2003) paper is cited and important findings from this paper are included in the revised manuscript.

- *page 2, line 34 - page 3, line 2: reading is lost due to large bracket text*

  AGREED AND CORRECTED.

- ***4.2 COSMO Model***

  *IMO the section 'COSMO model' should be divided into '2.1. General description' and '2.2. Parameterization of Convection'*
  AGREED AND IMPLEMENTED. We have re-structured our sections accordingly.

- *page 3, line 22: 'The equations are solved numerically on a Arakawa C-grid (Baldauf,2011)' - this is all you need here. Everything else would be too complicated.*
  AGREED AND IMPLEMENTED. Complicated stuff is omitted.

- *page 3, line 22: 'The temporal integration of the governing equation is done with' ...*

  AGREED AND CORRECTED.

- *page 3, line 23-24: Reformulate the sentence with the vertical layers. Please skip the number 50, because you are later on explaining the model configuration.*

  AGREED AND CORRECTED.

- *page 3, line 27-28: Please skip the sub-clause about diagnostic variables. This would be a list without end.*

  AGREED AND SKIPPED.

- *page 3, line 30-31: 'formation of precipitation fields' is a little bit too misleading. I would recommend to use 'The formation and modification of clouds and precipitating constituents'.*

  AGREED AND CORRECTED.

- *page 4, line 2-3: The sentence about Tiedtke can be skipped. The section 2.2. is discussing all details about moisture convection.*

  AGREED AND CORRECTED. This sentence is rephrased and details of Tiedtke scheme is skipped.

- *page 4, line 14-18: The sentence is too long.*

  AGREED AND REPHRASED.

- *page 5, line 14-18: The first two sentences should be shifted to section 3. The last sentence should be placed in section 2.1 (I suggested).*

  AGREED. These sentences are moved to section 2.1.

- ***4.3 Methods and Data***

  *This section should be rearranged. At first, a renaming to 'Methods' and 'Data' would be beneficial. Second, a good naming of section 3.1. is IMO 'Configuration of the Model simulations'. Third, the recent Section 4 should be Section 3.2. named 'Sensitivity experiments with NWP model'. Fourth, the recent section 3.2. should be Section 3.3. 'Observations'.*

  AGREED AND IMPLEMENTED. Section names are modified accordingly.

- *Regarding the COSMO model, it is needed to explicitly tell the version number. Having this version number, the community exactly knows about bugfixes and the state of research with your model version.*

  AGREED AND INCLUDED.

- *page 5, line 21: You do not explain 'VSCS'. I think it is very service convective storm.*

  NO CHANGES ARE DONE. VSCS refers to "Very Severe Convective Storm" and is mentioned in the Introductory section.

- *page 5, line 26+27: Two commas would be helpful after 'km' and 'latitudes'.*

  AGREED AND CORRECTED.

- *page 5, line 30-31: I do not understand this last sentence here.*

  AGREED AND REPHRASED.

- *page 6, line 2: ERA data are not an observation. It is a model forced to the atmospheric state observed. This is fully different than an observation. You can call it a reanalysis. Not more like this.*

  AGREED AND CORRECTED.

- *page 6, line 10-19: This paragraph should be shifted to Section 3.1. 'Configuration of model simulations'.*

  AGREED AND IMPLEMENTED.

- *page 7, line 2-3: Please skip everything starting from 'respectively'. You have already explained about that detail.*

  AGREED AND SKIPPED.

- *page 7, line 5-12: I never read before something complicated like this. Please re-formulate that paragraph in such a way that it is clear 'only the resolution changes compared to CPC.*

AGREED AND REPHRASED.

- *page 7, line 19-20: This last sub-clause is redundant information. You have already explained that for the other configurations.*

AGREED AND ELIMINATED. Redundant information is eliminated.

- ***4.4 Results and Discussions***

*IMO, this section consists of two subsections 4.1. and 4.2. The discussion about the location of the storm beginning at line 21 on page 11 is worth to put in an own subsection 4.3.*

AGREED AND IMPLEMENTED.

- *page 8: Where are the paragraphs here? One suggestion from my side: line 23.*

AGREED AND RESTRUCTURED THE NEW PARAGRAPH.

- *page 8, line 20-23: This deviation of the path is fully misleading here. The location of the storm is discussed later and needs in my opinion an own section.*

AGREED. We have revised this part, and introduced the discussion about location of the storm in appropriate places.

- *page 8, line 27: I cannot observe from Figure 3 that the magnitude of rainfall is larger for ERA, but the spatial extend of regions with a high amounts of rainfall is much larger in ERA than in COSMO. Furthermore, I see a shift in the maximum precipitation field between ERA and COSMO.*

AGREED. We have rephrased this part to bring more clarity. Important observations obtained from Figure are rephrased for better clarity.

- *page 9, line 10: Is this an observation from a radiosonde? If so, it should be highlighted here because then the reader knows which value is realistic (and not only a model output).*

AGREED AND INCLUDED. We have revised this sentence to include details.

- *page 9, line 12-13: You mention that the eye in COSMO forecast is 40 km away from observations. Yes, but the ERA is much far away from the observations. IMO, this discussion should be placed in 4.3. Otherwise, the information falls from the sky.*

AGREED. We have taken care of this sudden jump in the revised manuscript and appropriate changes are done.

- *page 9, line 24-30: I miss the discussion about the placement of the CAPE maximum at Figure 4. It is evident that the runs with 24 lead times place the maximum more the South compared to the runs with longer lead times.*

AGREED AND CORRECTED.

- *page 9, line 32-33: You argue something about downdrafts and updrafts, but no figure or detailed text is given. What do you exactly mean? What is a realistic downdraft and updraft? Such a discussion would be a chance to improve the paper and make it more scientific.*

AGREED. We have included more details with a new figure to discuss the vertical structure of cyclonic storm.

- *page 10, line 12: ERA is not observations. Please skip that.*

AGREED AND SKIPPED.

- *page 10, line 12-13: You argue that the CAPE values of the ERA are always overestimated, but you do not give a proof for it. IMO, this sentence can be*

*skipped.*

AGREED AND SKIPPED.

- *page 10, line 15-23: There are no observation by the IMD shown. Thus, the reader has no feeling for the differences between model and observations.*

  AGREED. We have included IMERG satellite-based precipitation measurements.

- *page 10, line 27: the ERA reanalysis fields show an overestimation in spatial extend but without any observations the reader would not believe that magnitudes of precipitation and CAPE are overestimated.*

  AGREED AND CORRECTED.

- *page 11, line 2-3: I do not understand the meaning of that sentence.*

  AGREED AND CORRECTED.

- *page 11, line 5: This is a barplot and not a histogram.*

  AGREED AND CORRECTED.

- *page 11, line 7-9: A sentence without content. Please skip it.*

  AGREED AND SKIPPED.

- *page 11, line 9-12: The discussion about leadtime requirements is confined to precipitation intensities but not to location of intense precipitation. I do not understand, why this is less important. Regarding lead times, this is a crucial point.*

  AGREED. We have included more details and discussed this aspects in Results and Discussion.

- *page 11, line 19-20: Again, as already said, what is the value of such sentence without having seen any observation.*

AGREED AND SKIPPED.

- *page 11, line 31-35: The critical discussion about predictability only includes the model domain. However, the quality of the initial and lateral boundary conditions is of much more importance, but it is not discussed and analyzed at all.*

  AGREED AND CORRECTED.

- ***4.5 Conclusions***

  *The conclusions are too general and off-topic. The main content is about preconditions for high-resolution simulations and improvements or problems detected in other studies. The relation to this paper is not so clear. IMO, the conclusion should be rewritten in order to get a clue about the implications of the author for the whole scientific community.*

  AGREED AND RE-WRITTEN.

- *page 12, line 9-12: Too long sentence.*

  AGREED AND REPHRASED.

- *page 12, line 15-17: What is the measure that indicates deep convection on 3rd of December 2017?*

  DETAILS INCLUDED. Large values of CAPE and associated accumulated rainfall over the location was the important aspects concerned with the deep convection. We have addressed this point in the revised manuscript.

- *page 12, line 25-29: What is the line of argumentation here? The text deals initially with COSMO-DE and its graupel scheme. Afterwards, we learned something about reduced precipitation over the coastal Arabian and then, downscaling issues of the UM are referred. There is no logic at all.*

  NECESSARY CORRECTIONS ARE DONE, AND SENTENCE IS REPHRASED.
- *page 12, line 33-34: What do you mean with that sentence? What means necessary?*

  NECESSARY CORRECTIONS ARE DONE, AND SENTENCE IS REPHRASED.

- *page 13, line 4-5: The english text reads strange starting from 'where little ...'.*

  NECESSARY CORRECTIONS ARE DONE, AND SENTENCE IS REPHRASED.

- *page 13, line 6-8: The author is telling about tuned parameterizations in NWP models, in particular for specific resolutions and scales. The study presented here should give valuable insight into the treatment of convection and the impact on precipitation. I am not convinced at all that we learn with this study something new and not known from former studies. We learn about model results from the storm 'OCKHI' nothing more. This study does not help to conclude about what are the problems with dynamical downscaling nor at which resolution to switch off parts of the parameterization of convection.*

  SUBSTANTIAL REVISION IS DONE, AND CONVINCING CONCLUSIONS ARE INCLUDED.

- ***4.6 Figures and Tables***

  *figure 1: Do we need it? The text explains everything one needs.*
  PARTLY AGREED. We are still retaining this figure. However, we have modified this figure for making it more informative.

- *figure 2: Which simulation is shown regarding the CAPE? The extend of the COSMO domain is not large enough to extract that information.*

  AGREED. As we have extended our model domain, this aspect is addressed.

- *figure 3: Which experiment of COSMO is shown? (CPC)*

NECESSARY CORRECTIONS ARE DONE AND DETAILS ARE INCLUDED IN THE FIGURE CAPTION.

- *figure 4: The CAPE observation at 00 UTC of 3.12.2017 and at 69.15 degree East and 11.82 degree North should be marked in each plot.*

  NECESSARY DETAILS ARE INCLUDED IN THE REVISED MANUSCRIPT.

- *figure 5: Which area is taken for averaging?*

  NECESSARY DETAILS ARE INCLUDED IN THE REVISED MANUSCRIPT.

- *figure 6: Which area is taken for averaging? What is meant with the last sentence in the caption?*

  SENTENCE IS REPHRASED AND OTHER DETAILS ARE INCLUDED IN THE REVISED MANUSCRIPT.

- *Table A1: The version number of COSMO is missing. Reference for grid-scale precipitation, vertical turbulence diffusion and surface-layer turbulent fluxes is missing.*

  NECESSARY CITATIONS ARE INCLUDED IN THE REVISED TABLE.

- *Table A2: Which time is analyzed? The position at 00UTC of 3.12.2017? Why is the analysis not done for other stages of the storm?*

  NECESSARY DETAILS ARE INCLUDED IN THE REVISED MANUSCRIPT.